**RESEARCH**

# Gene count normalization in single-cell imaging-based spatially resolved transcriptomics

Lyla Atta[1,2], Kalen Clifton[1,2], Manjari Anant[2,3], Gohta Aihara[1,2] and Jean Fan[1,2]*

*Correspondence:
jeanfan@jhu.edu

[1] Department of Biomedical
Engineering, Johns Hopkins
University, Baltimore, MD 21218,
USA
[2] Center for Computational
Biology, Whiting School
of Engineering, Johns Hopkins
University, Baltimore, MD 21211,
USA
[3] Department of Neuroscience,
Johns Hopkins University,
Baltimore, MD 21218, USA

## Abstract

**Background:** Recent advances in imaging-based spatially resolved transcriptomics (im-SRT) technologies now enable high-throughput profiling of targeted genes and their locations in fixed tissues. Normalization of gene expression data is often needed to account for technical factors that may confound underlying biological signals.

**Results:** Here, we investigate the potential impact of different gene count normalization methods with different targeted gene panels in the analysis and interpretation of im-SRT data. Using different simulated gene panels that overrepresent genes expressed in specific tissue regions or cell types, we demonstrate how normalization methods based on detected gene counts per cell differentially impact normalized gene expression magnitudes in a region- or cell type-specific manner. We show that these normalization-induced effects may reduce the reliability of downstream analyses including differential gene expression, gene fold change, and spatially variable gene analysis, introducing false positive and false negative results when compared to results obtained from gene panels that are more representative of the gene expression of the tissue's component cell types. These effects are not observed with normalization approaches that do not use detected gene counts for gene expression magnitude adjustment, such as with cell volume or cell area normalization.

**Conclusions:** We recommend using non-gene count-based normalization approaches when feasible and evaluating gene panel representativeness before using gene count-based normalization methods if necessary. Overall, we caution that the choice of normalization method and gene panel may impact the biological interpretation of the im-SRT data.

**Keywords:** Normalization, Scaling factor, Spatial transcriptomics, Differential expression

## Background

Imaging-based spatially resolved transcriptomics (im-SRT) represent a new set of technologies that enable the high-throughput detection of targeted RNA species in fixed tissues [1–3]. While im-SRT data is generally acquired at a single-molecule resolution, it is often aggregated into single-cell resolution by counting RNA molecules within cell regions identified by cell segmentation [4–7]. This achieves single-cell resolution spatially resolved gene expression profiling to enable the interrogation of cellular function and organization within tissues to enhance our understanding of cellular spatial microenvironments in biological processes such as tissue development, disease progression, and drug response [1, 2, 8].

In order to discover biological insights, normalization of gene expression data is often needed to account for technical differences that may confound underlying biological signals [4, 9, 10]. Common normalization methods for gene expression count data include scaling by a sample's total detected gene counts (also called size factor normalization, relative counts normalization, and library size normalization). This normalization approach is commonly used in the analysis of single-cell RNA-sequencing (scRNA-seq) data to account for technical differences in RNA capture rates between cells and has also been applied to im-SRT data [11, 12]. Alternative normalization methods such as scTransform, DESeq2, and TMM consider differences in gene expression composition between samples and can also be applied to im-SRT data though scTransform was designed for scRNA-seq while DESeq2 and TMM were designed for bulk RNA-seq data [9–13]. Specific to im-SRT data, cell volume normalization has been previously applied to account for partially imaged cell volumes [14–16].

Given the capacities of current im-SRT technologies, a few hundred unique gene species can typically be measured simultaneously in a single experiment. Because gene detection probes are designed based on a target gene's sequence, these gene species must be pre-selected as part of a gene panel in the experimental design [2, 4]. These gene panels may be designed with the intention of spatially profiling specific cell types or biological processes of interest and may thus include genes that are known to be highly expressed and/or are canonical cell type or pathway markers based on domain knowledge and/or as informed by scRNA-seq [14–18]. In recent years, commercial options for im-SRT have become available, where off-the-shelf gene panels with probes for a pre-selected set of genes can be used. These gene panels may focus on specific cell types such as immune cells or neurons, or specific pathways and processes [19]. As such, these gene panels may be skewed to enable more thorough characterization of the cell types of interest at the expense of other cell types present in the tissue.

Here, we investigate how different gene count normalization procedures may impact downstream transcriptomic analyses for im-SRT data with different gene panels (Fig. 1). Ideally, biological conclusions drawn from im-SRT studies in the same biological system would be robust to experimental design choices such as gene panel composition. For example, if a gene is differentially expressed in a particular tissue region or cell type in a given biological system probed with one gene panel, it should also be differentially expressed at a similar fold change given a different gene panel, if that the gene is in both panels. Similarly, if a gene is identified as having spatially

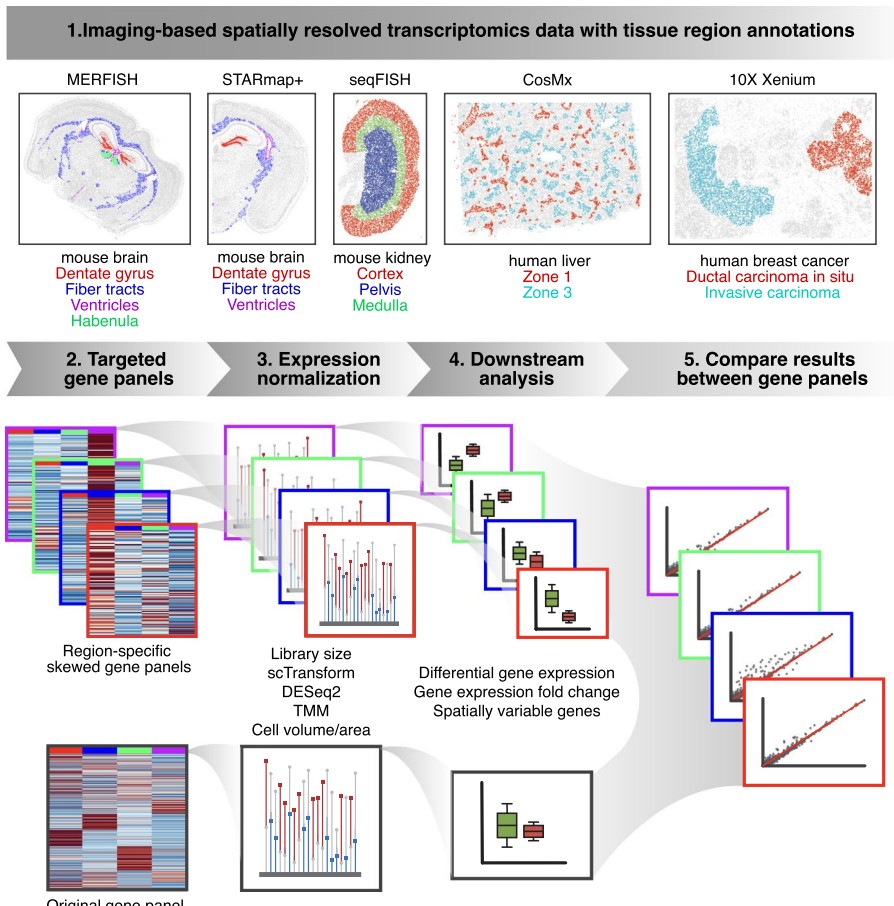

**Fig. 1** im-SRT normalization evaluation workflow. (1) 5 publicly available im-SRT datasets representing a variety of tissues and im-SRT technologies. Cells colored by tissue region annotations. (2) im-SRT datasets with skewed gene panels were simulated by sampling genes overrepresenting different tissue regions. (3) Up to 5 gene count normalization approaches were applied to each im-SRT dataset with the simulated skewed gene panels as well as with the original full gene panel. (4) Downstream analysis of spatial gene expression data including differential gene expression, gene expression fold change, and spatially variable gene analysis were performed. (5) Ideally, biological conclusions would be robust to gene panel choice. Robustness of downstream analysis results after each normalization approach was investigated by comparing results for each im-SRT dataset with each skewed gene panel compared to the corresponding full gene panel

variable gene expression in one gene panel, it should also be identified to be spatially variable in a different gene panel. However, we show using im-SRT data with simulated skewed gene panels that different normalization methods may introduce errors in downstream analyses such as identifying significantly differentially expressed genes, analyzing gene expression fold changes, and identifying spatially variable genes. We further demonstrate that these concerns generalize across diverse im-SRT technologies and tissues. Finally, using simulated gene panels of different sizes, we find that these concerns can be mitigated as gene panels increase in size. Overall, we caution that the choice of normalization method and gene panel may impact the biological interpretation of the im-SRT data, and present readers with a decision tree to help guide the choice of gene expression normalization method for their im-SRT data.

## Results

To investigate the potential impact of normalizing im-SRT data with different gene panels, we simulated im-SRT datasets with different gene panels. We first focused on a multiplexed error-robust fluorescent in situ hybridization (MERFISH) im-SRT dataset of a coronal section of the mouse brain that originally profiled 483 genes representative of diverse cell types (Additional file 1: Fig. S1A) [5, 20].

As a proof of principle, we created different gene panels that could allow us to better characterize different mouse brain regions, specifically the ventricles, habenula, fiber tracts, and dentate gyrus (Additional file 1: Fig. S1B-E). Many brain regions are enriched for specific cell types and in turn differentially express specific genes when compared to other brain regions. To achieve this, we annotated mouse brain regions by aligning and transferring region annotations from the Allen Brain Atlas using STalign [21, 22]. Then, for a brain region of interest, we randomly sample 100 genes that are significantly overexpressed in this region, thereby simulating a 100-gene gene panel skewed to our brain region of interest. We quantify gene panel skewness using a Kullback–Leibler divergence-based metric to confirm that the original full gene panel as well as random 100-gene gene panel are more representative of all brain regions compared to simulated region-skewed gene panels (Additional file 1: Fig. S1F-G, methods). For both the original full gene panel and the region-skewed panel, we compared library size, scTransform, DESeq2, TMM, and cell volume normalization, in addition to no normalization.

### Skewed gene panels with different normalization methods result in region-specific effects on scaling factors and normalized gene expression magnitudes in im-SRT data

To investigate the potential impact of normalization procedures on downstream transcriptomic analyses, we first compared the scaling factors derived from the full and ventricle-skewed gene panel for each normalization method (Fig. 2A). Given that these scaling factors are applied to detected cell gene counts to obtain the normalized gene expression magnitudes used in downstream analysis, we expect that differences in the scaling factors between gene panels will result in differences in the normalized gene expression magnitudes, which may in turn impact downstream analyses. For normalization methods such as library size, DESeq2, and TMM where scaling factors are derived based on gene count information, we further expect that the scaling factors will differ depending on the gene panel used. We note that scTransform does not use cell-specific scaling factors and was therefore omitted from this analysis.

Under library size normalization, when we compared scaling factors with the ventricle-skewed gene panel to the full gene panel, we found that scaling factors for cells in the ventricle region tended to be systematically larger than for cells in other regions. This is expected because cells in the ventricle region, by design, have higher expression of genes in the ventricle-skewed gene panel than cells in other regions (Additional file 1: Fig. S1B). In contrast, with the full gene panel that is more representative of gene expression of all analyzed brain regions, cells in the ventricle region do not have systematically larger scaling factors compared to cells in other regions. We also see this region-specific effect with DESeq2 and TMM normalizations (Fig. 2A). We quantified the root mean squared error (RMSE) in scaling factors after using different normalization approaches

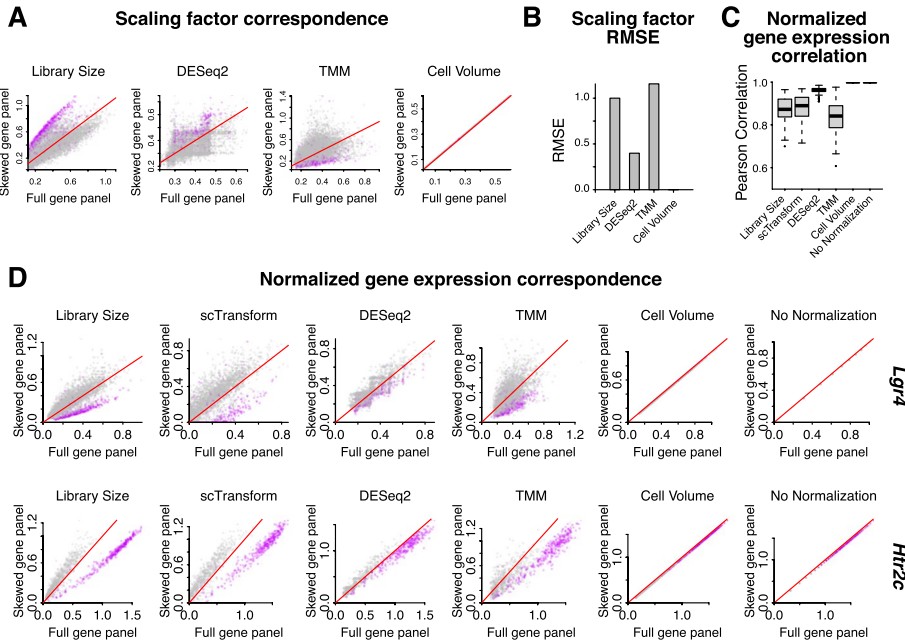

**Fig. 2** Impact of region-skewed gene panel on normalization scaling factors and normalized gene expression magnitudes. **A** For a coronal section of the mouse brain assayed by MERFISH, scatterplots of $\log_{10}$(scaling factors) for each cell based on different normalization methods (library size normalization, DESeq2 normalization, TMM normalization, and cell volume normalization) for the ventricle-skewed gene panel versus the full gene panel. Cells within the ventricle brain region are shown in purple. Cells in the fiber tract, habenula, and dentate gyrus brain regions are in grey. Red line indicates $x = y$. **B** Root mean squared error in scaling factors after each normalization method between the ventricle region-skewed gene panels and the full gene panel. **C** Boxplot of Pearson correlation coefficients (*r*) across genes for normalized gene expression magnitudes with the full gene panel versus the ventricle region-skewed gene panel across different normalization methods. **D** Scatterplot of normalized gene expression magnitudes when normalization was performed with the full gene panel versus the ventricle skewed gene panel for two highly expressed genes, *Lgr4* and Htr2c. Cells within the ventricle brain region are shown in purple. Cells in the fiber tract, habenula, and dentate gyrus regions are in grey. Red line indicates $x = y$

with the ventricle-skewed gene panel compared to the full panel (Fig. 2B). We find that these results are consistent for gene panels designed for other brain regions including the habenula, fiber tracts, and dentate gyrus (Additional file 1: Fig. S2A-D). Similar to the ventricle region, these other brain regions are enriched for specific cell types and in turn differentially express specific genes when compared to other brain regions. As such, differences in scaling factors for regions outside of those to which a gene panel is skewed are driven by the underlying differences in cell type composition and resulting gene expression differences. These results indicate that in im-SRT data with region-skewed gene panels, normalization using scaling factors derived based on gene counts may introduce region-specific impacts on downstream analyses depending on the normalization procedure used. Alternatively, as cell volume is independent of the number of gene counts detected in a cell, the scaling factor is the same with the skewed and full gene panels and therefore does not result in region-specific impacts on downstream analyses.

Next, we performed gene count normalization with each of the normalization methods. We first assessed how consistent normalized gene expression magnitudes are when using the ventricle-skewed gene panel compared to normalized gene expression magnitudes with the full gene panel. To do this, we computed the correlation in normalized

gene expression magnitudes between the skewed panel and the full gene panel (Fig. 2C). Normalized gene expression magnitude correlations are the lowest for TMM normalization, ranging from around 0.5 to 0.98 for different genes. Library size and scTransform normalization resulted in similarly low correlations while DESeq2 resulted in comparatively higher correlations, all above 0.9. Volume normalization and no normalization resulted in perfectly correlated normalized gene expression magnitudes, as expected given that scaling factors are unchanged between different gene panels. We again find that these results are consistent for gene panels skewed towards other brain regions (Additional file 1: Fig. S2E).

To further understand the source of differences in normalized gene expression magnitude correlations between skewed and full gene panels with different normalization methods, we looked at the normalized gene expression magnitudes for select highly expressed genes (Fig. 2D). Notably, given a gene panel skewed towards genes enriched in the ventricle region, under library normalization, normalized gene expression magnitudes for cells in the ventricle region are smaller than with the full gene panel, as expected given the relatively larger scaling factors. We observe similar effects with scTransform. We note that while scTransform does not use cell-specific scaling factors, it performs gene-specific adjustments of expression magnitude by regressing out the effect of library size from observed counts using a generalized linear model framework [13]. Since this is in effect correcting for cell library size, we expect to observe similar effects to those observed with library size normalization. For other regions, normalized gene expression magnitudes are higher with the skewed gene panel compared to the full gene panel. This is consistent with the smaller scaling factors for cells in these regions with the skewed gene panel compared to the full gene panel. As with the scaling factors, region-specific differences between normalized counts in skewed and full gene panels are less pronounced with the DESeq2 and TMM normalizations than with library normalization. These examples show that the region-specific impacts on scaling factors when using region-skewed gene panels with count-based normalizations can in turn differentially impact gene expression magnitudes in a region-specific manner resulting in distortions in the relationship between normalized gene expression magnitudes across cells.

### Skewed gene panels with different normalization methods may lead to biases in differential gene expression results

Next, to evaluate the impact of these different normalization approaches with different gene panels on downstream analysis, we performed differential gene expression and gene fold change analysis. To do this, we compared the *p*-value of a Wilcoxon rank sum test of differential expression as well as the $\log_2$ average fold change between cells in each brain region and cells in all other regions based on normalized gene expression magnitudes achieved with the skewed gene panel versus the full gene panel for each normalization method. When using library size and scTransform normalization, *p*-values for ventricle-vs-all differential gene expression tests have less significant *p*-values with the ventricle-skewed gene panel compared to the full gene panel. In contrast, other differential gene expression test *p*-values are similar when using the ventricle-skewed gene panel and the full gene panel. This region-specific difference is less pronounced with

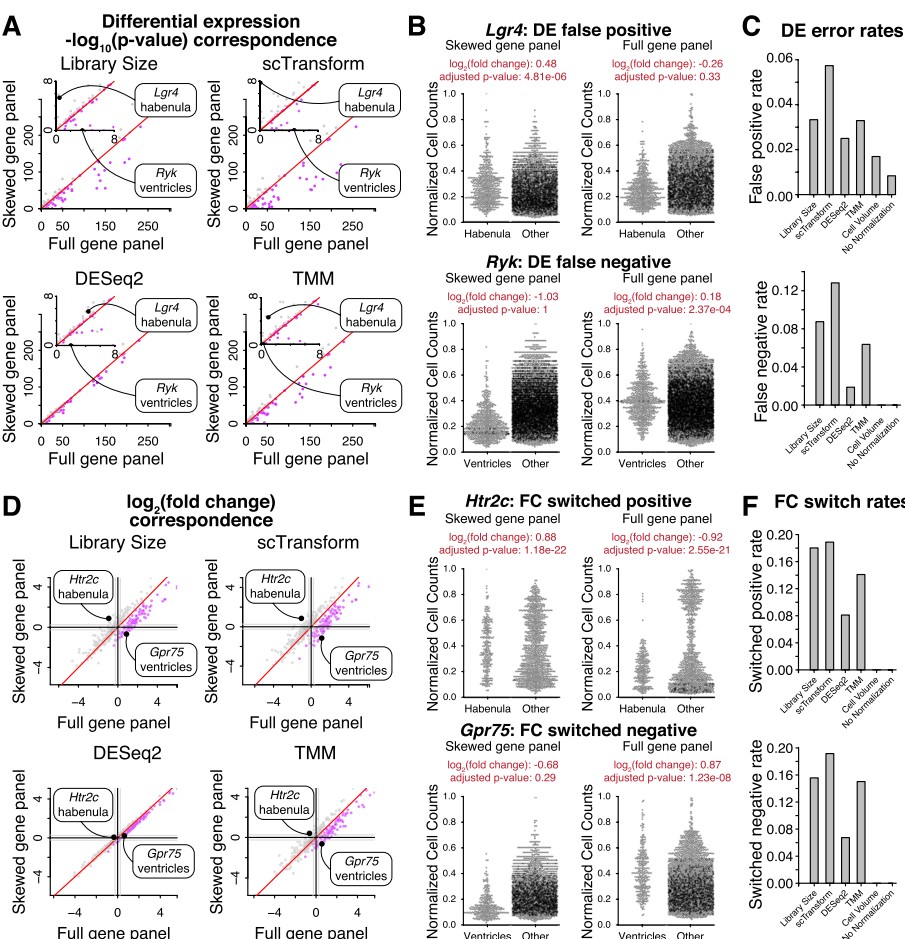

**Fig. 3** Impact of region-skewed gene panel on differential expression analysis and fold change evaluation. **A** Wilcoxon rank sum test $-\log_{10}$($p$-values) adjusted for multiple hypothesis testing for one region-vs-all differential gene expression test with normalized gene expression from the full gene panel versus the ventricle-skewed gene panel. Red line indicates $x = y$. Inset shows smaller $-\log_{10}$(adjusted $p$-values). Purple indicates ventricle-vs-all tests, and grey indicates other region-vs-all tests. **B** Library size normalized gene expression magnitudes in cells in region of interest vs cells in other regions with the full gene panel and the ventricle skewed gene panel for the false positive (*Lgr4,* top) and false negative (*Ryk,* bottom) genes highlighted in **A**. **C** Differential gene expression false positive (top) and false negative (bottom) rates after each normalization method, taking differential gene expression results with the full gene panel to be ground truth. **D** One region-vs-all gene fold change with normalized gene expression from the full gene panel and the ventricle gene panel. Black lines indicate 0 $\log_2$ fold change and grey lines indicate 0.5 $\log_2$ fold change. Purple indicates ventricle-vs-all comparisons, grey indicates other region-vs-all comparisons. **E** Library size normalized gene expression magnitudes in cells in region of interest vs cells in other regions with the full gene panel and the ventricle skewed gene panel for the fold change switched positive (*Htr2c,* top) and switched negative (*Gpr75,* bottom) genes highlighted in **D**. **F** Gene expression fold change switched positive (top) and switched negative (bottom) rates after each normalization method, taking fold change results with the full gene panel to be ground truth

TMM and DESeq2 normalization (Fig. 3A). Note that with no normalization and volume normalization, the normalized gene expression magnitudes are the same (Fig. 2) and thus downstream analysis results between the skewed gene panel and the full gene panel will be identical.

To understand the source of differences in differential gene expression testing results after library size normalization, we looked at the normalized gene expression

magnitudes for individual highly expressed genes where the result of differential gene expression testing is not concordant between skewed and full gene panels (Fig. 3B). As examples, for the gene *Lgr4*, library size normalization with the ventricle-skewed gene panel results in normalized gene expression magnitudes that are on average slightly larger in the habenula region compared to other regions. In contrast, library size normalization with the full gene panel results in normalized gene expression magnitudes that are smaller in the habenula region compared to other regions. This results in a false positive on a one-sided Wilcoxon rank sum test of differential expression between cells in the habenula region and cells in all other regions, taking the result with the full gene panel to be ground truth. Conversely, for the gene *Ryk*, library size normalization with the ventricle-skewed gene panel results in normalized gene expression magnitudes that are much smaller in the ventricle region compared to other regions. In contrast, library size normalization with the full gene panel results in normalized gene expression magnitudes that are similar between the ventricle region and other regions. This results in a false negative on a one-sided Wilcoxon rank sum test of differential expression between cells in the ventricle region and cells in all other regions, taking the result with the full gene panel to be ground truth. These region-specific impacts are consistent with the differences in library size scaling factors resulting from the skewed gene panel, where cells in the ventricle region have larger scaling factors than those in other regions because of their higher expression of genes included in the gene panel. We quantify the rates of false positive and false negative differential expression test results (Fig. 3C) and find up to a 13% error rate with multiple hypothesis correction depending on the count-based normalization approach used.

With respect to gene expression fold changes, under library normalization and scTransform, $\log_2$ fold changes in ventricle-vs-all comparisons have lower $\log_2$ fold changes when using the ventricle-skewed gene panel compared to the full gene panel with some genes having negative $\log_2$ fold changes in the skewed gene panel and positive $\log_2$ fold changes in the full gene panel (switched negative). Conversely, $\log_2$ fold changes in comparisons for other regions (habenula-, fiber tract-, and dentate gyrus-vs-all) have more similar fold changes but include some genes that have a positive $\log_2$ fold change in the skewed gene panel and a negative $\log_2$ fold change in the full gene panel (switched positive). These region-specific effects are also seen with TMM normalization but are less pronounced and are not seen with DESeq2 normalization (Fig. 3D).

To understand the source of differences in $\log_2$ fold changes between skewed and full gene panels after library size normalization, we looked at the normalized gene expression magnitudes for individual highly expressed genes where $\log_2$ fold changes are not concordant between skewed and full panels (Fig. 3E). As examples, for the gene *Htr2c*, library size normalization with the ventricle-skewed gene panel results in normalized gene expression magnitudes in the habenula region that are more similar in magnitude to those in other regions. In contrast, library size normalization with the full gene panel results in normalized gene expression magnitudes that are lower in the habenula region compared to other regions in the data. This results in a switched positive $\log_2$ fold change with a positive $\log_2$ fold change with the ventricle-skewed gene panel compared to a negative $\log_2$ fold change with the full gene panel, with similar absolute fold change magnitudes and significant adjusted *p*-values. Conversely, we observe that for the gene

*Gpr75*, library size normalization with the ventricle skewed gene panel results in normalized gene expression magnitudes in the ventricle region that are lower compared to other regions. In contrast, library size normalization with the full gene panel results in normalized gene expression magnitudes that are smaller in the ventricle region compared to other regions, resulting in a switched negative $\log_2$ fold change. We quantify the rates of switched positive and switched negative fold change results (Fig. 3F) and find up to a 19% switch rate depending on the count-based normalization approach used.

These examples show that the region-specific impacts on scaling factors when using region-skewed gene panels with count-based normalization can result in distortions in normalized gene expression distributions, which in turn lead to inconsistent fold change and differential gene expression results with different gene panels. Again, results are consistent for gene panels designed for other brain regions with the MERFISH data (Additional file 1: Fig. S3-4).

To further investigate if these observations extend to im-SRT data acquired by other technologies and from other tissues, we performed the same analyses with several other im-SRT datasets. We evaluated the effects of different normalization approaches with simulated skewed gene panels in STARmap PLUS [23] data from mouse brain, seqFISH data from mouse kidney [24], CosMx data from human liver [25], and 10X Xenium data from human breast cancer [26] (Additional file 1: Fig. S5-8). Skewed gene panels were simulated by sampling differentially expressed genes for regions identified manually based on tissue architecture (seqFISH mouse kidney), from author provided pathology annotations (STARmap PLUS mouse brain, 10X Xenium human breast cancer), or from author provided annotations derived from spatial-cell type clustering (CosMx human liver). We observe similar region-specific effects on scaling factors, differential gene expression *p*-values, and gene $\log_2$ fold changes across im-SRT technologies and tissues. We quantify differential expression testing error rates and fold change switch rates (Fig. 4). While error rates vary by dataset and simulated gene panel, we generally observe higher differential expression error rates with library size and scTransform normalization with skewed gene panels, reaching up to 30% with some simulated gene panels. Similarly, we generally observe higher fold change switch rates with library size and scTransform normalization reaching up to 60% with some simulated gene panels. We note that we observe higher rates of fold change switching at more modest fold changes compared to at higher fold changes where fold change directions tend to remain consistent between skewed gene panels and their corresponding full gene panel.

Overall, these results show that the region-specific effects of gene count-based normalization in im-SRT with skewed gene panels may result in false positives and negatives in differential gene expression testing, as well as fold changes with opposite directions. Further, these results suggest that region-specific differences in transcriptomics analysis results using skewed gene panels also generalize to im-SRT data from other imaging-based spatial transcriptomics technologies.

### Skewed gene panels with different normalization methods may lead to false negatives in spatially variable gene expression analysis

To investigate the impact of different normalization methods with skewed gene panels on downstream analyses specific to SRT data, we sought to identify spatially variable

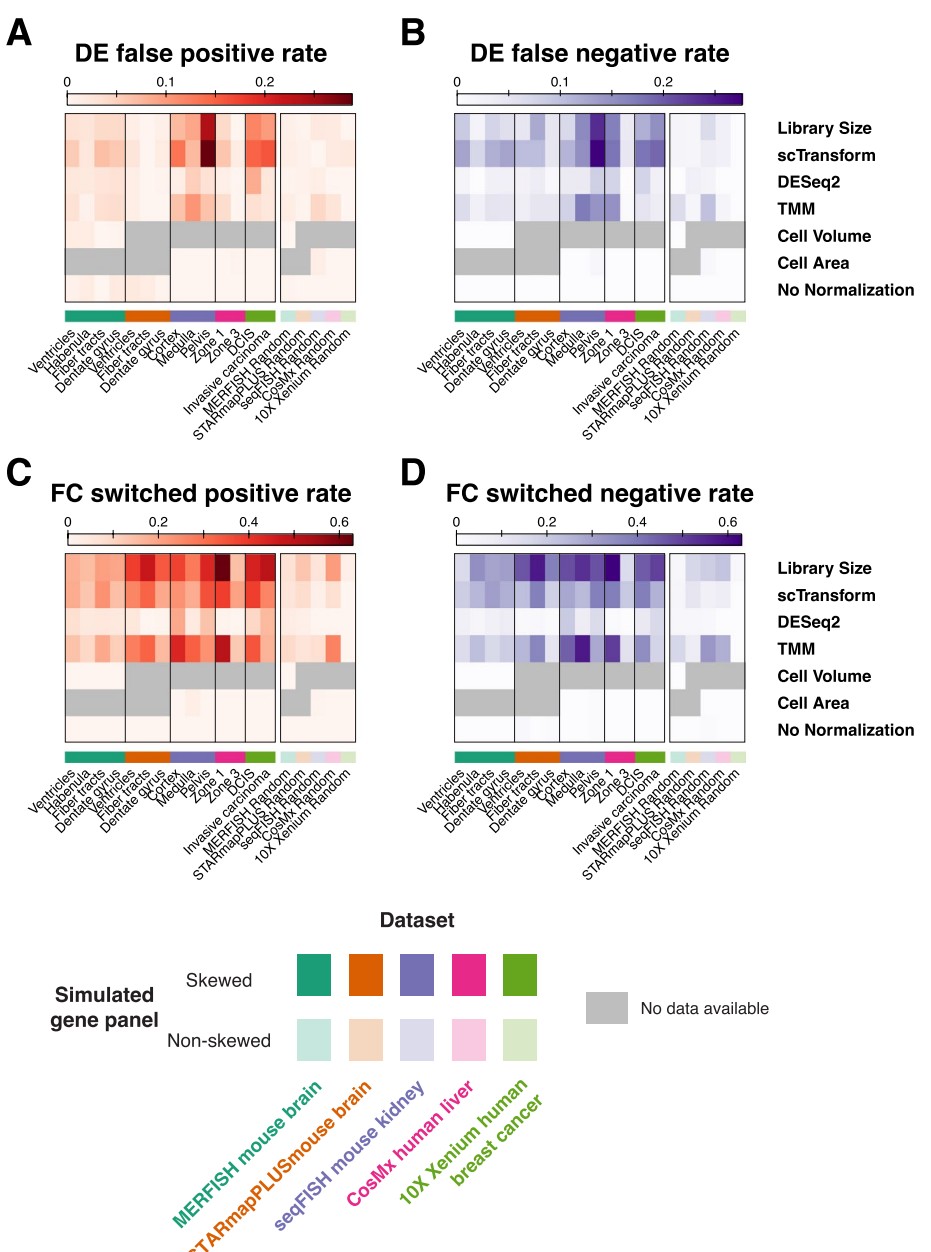

**Fig. 4** Impact of normalization and region-skewed and non-skewed gene panels on differential expression analysis and fold change evaluation for multiple simulated gene panels in 5 analyzed im-SRT datasets. **A** Differential gene expression false positive rates after each normalization method with each simulated gene panel. **B** Differential gene expression false negative rates after each normalization method with each simulated gene panel. **C** Gene expression fold change switched positive rates after each normalization method with each simulated gene panel. **D** Gene expression fold change switched negative rates after each normalization method with each simulated gene panel

genes (SVGs), genes with highly spatially correlated expression patterns. We used nnSVG with SEraster preprocessing to identify SVGs after gene count normalization with the ventricle-skewed gene panel and compared the results to those obtained with the full gene panel (Fig. 5A) [27, 28]. Comparing *p*-values with the ventricle-skewed

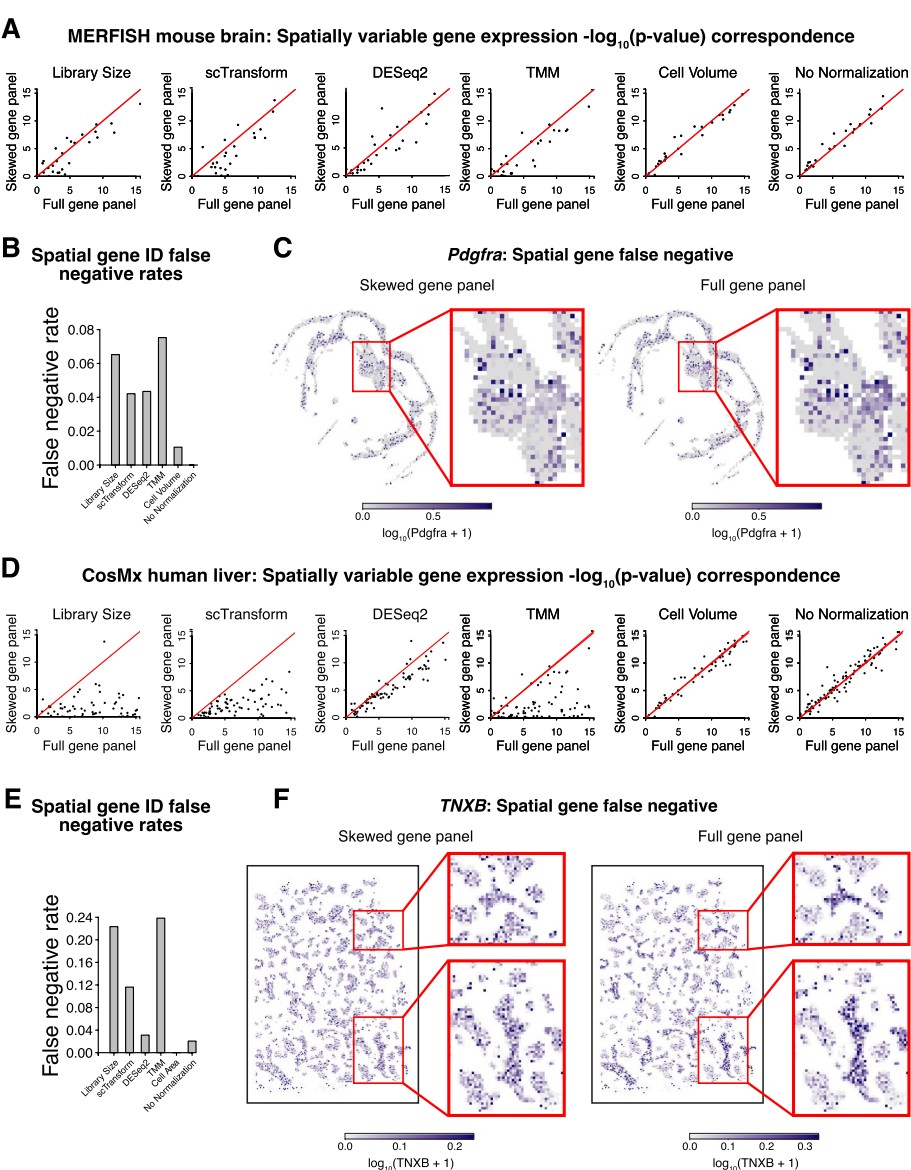

**Fig. 5** Impact of region-skewed gene panel on spatially variable gene identification. **A** nnSVG significant spatially variable gene expression test $-\log_{10}(p$-values) adjusted for multiple hypothesis testing with normalized gene expression from the full gene panel versus the ventricle-skewed gene panel. Red line indicates $x = y$. **B** Significant spatially variable gene expression false negative rates after each normalization method, taking significant spatially variable gene expression results with the full gene panel to be ground truth. **C** Rasterized (50 μm) library size normalized gene expression magnitudes visualized in tissue space for *Pdgfra*, a representative spatial gene false negative, with the skewed gene panel (left) and the full gene panel (right). **D** Spatially variable gene identification in CosMx human liver im-SRT data with simulated Zone 1 region-skewed gene panel and full gene panel. nnSVG significant spatially variable gene expression test $-\log_{10}(p$-values) adjusted for multiple hypothesis testing with normalized gene expression from the full gene panel versus the Zone 1-skewed gene panel. Red line indicates $x = y$. **E** Significant spatially variable gene expression false negative rates after each normalization method, taking significant spatially variable gene expression results with the full gene panel to be ground truth. **F** Rasterized (50 μm) library size normalized gene expression magnitudes visualized in tissue space for *TNXB*, a representative spatial gene false negative, with the skewed gene panel (left) and the full gene panel (right)

gene panel to those with the full gene panel, we find that while there is some discordance in *p*-values without normalization and with cell volume normalization, likely reflective of the stochasticity of the nnSVG algorithm, this discordance is more pronounced with the evaluated count-based normalization methods. We quantify this effect by computing the false negative rate of SVG identification and we find higher false negative rates across count-based normalization methods with the ventricle-skewed gene panel (Fig. 5B). As an example, we visualize the spatial gene expression pattern of one such false negative gene, *Pdgfra* (Fig. 5C). Library size normalized *Pdgfra* expression with the ventricle-skewed gene panel does not show a discernible spatial pattern, compared to with the full gene panel, where we observe a region of high normalized *Pdgfra* expression. We observe similar results with gene panels designed for other brain regions with the MERFISH data (Additional file 1: Fig. S9).

We further investigate the impact of different normalization methods with skewed gene panels on SVG analysis in im-SRT data of other tissues from different technologies. We observe similar effects in the CosMx data of human liver with the Zone 1-skewed gene panel. Comparing *p*-values with the Zone 1-skewed gene panel to those with the full gene panel, we find relatively higher levels of discordance with count-based normalization methods compared to with cell volume normalization or without normalization (Fig. 5D). As with the ventricle-skewed gene panel with the MERFISH mouse brain im-SRT data, this effect is evident in the higher false negative rates of SVGs identified across count-based normalization methods, reaching up to 24% (Fig. 5E). We can see an example of this effect when looking at the spatial gene expression pattern of one false negative gene, *TNXB* (Fig. 5F). Library size normalized *TNXB* expression with the Zone 1-skewed gene panel does not show a discernible spatial pattern, compared to with the full gene panel, where we observe several regions of high *TNXB* expression relative to the rest of the tissue.

We note that in several of the data sets explored in this analysis, nnSVG analysis finds most genes to be spatially variable with the original gene panels. This is likely due to the spatial organization and variability of transcriptionally distinct cell types. Since we treat genes with non-significant spatial variation with the full panel as true negatives in our error rate analysis, in these cases we find few or no true negatives and therefore omit quantification of false positive rates.

Overall, these results show that gene count-based normalization with skewed gene panels may result in unreliable results in downstream SVG analysis. Further, we observe these results across skewed gene panels and im-SRT datasets suggesting that these impacts on downstream SVG analyses can generalize to im-SRT data from different tissues assayed by different technologies.

### Normalization-induced biases can be mitigated with more representative or larger gene panels

Next, we investigated if impacts on the reliability of differential gene expression testing and fold change analysis resulting from region-skewed gene panels can be mitigated with gene panels that are more representative of the gene expression of component regions in the tissue. To do this, we simulated a non-skewed gene panel of the same size as the region-specific gene panels used in the previous simulations (Additional file 1: Fig.

 

S1F) and compared scaling factors, and results of differential expression testing and fold change analysis between the full gene panel and non-skewed gene panel after normalization with each normalization methods (Additional file 1: Fig. S10). We first computed scaling factors for each normalization method and compared to those computed with the full gene panel. In contrast to the skewed gene panel, scaling factors were generally similar between the skewed gene panel and the full gene panel and no region-specific differences between regions were observed (Additional file 1: Fig. S10A). This effect is quantified in the scaling factor RMSE, where the non-skewed gene panel generally has a lower RMSE across normalization methods compared to the region-skewed gene panels (Additional file 1: Fig. S10B). We note that TMM performs normalization by selecting a "reference" sample introducing a larger degree of variation in TMM results, compared to other normalization methods, reflective of the choice of reference sample. As before, scaling factors for volume normalization are independent of gene panel and therefore are the same between the non-skewed and full gene panels. Looking at correlation of normalized gene expression magnitudes with the non-skewed gene panel compared to the full gene panel, we find that correlations after library size, scTransform, and DESeq2 are generally above 0.9 (Additional file 1: Fig. S10C). We then performed differential expression testing and found that both *p*-values and fold changes were well correlated between the non-skewed and the full gene panels for all genes (Additional file 1: Fig. S10D,F). Quantifying differential gene expression false positive and false negative rates and gene fold change switched positive and switch negative rates, we find that the non-skewed gene panel generally has lower error rates compared to region-skewed gene panels (Additional file 1: Fig. S10E, G). Finally, we simulate non-skewed gene panels using other im-SRT data and find similarly low rates of differential gene expression false positives and false negatives and gene fold change switched positives and switch negatives (Fig. 4). These results indicate that region-specific impacts on differential gene expression testing and fold change analysis resulting from count-based normalization can be mitigated by designing gene panels that are more representative of the gene expression of component regions and cell types in the tissue.

We anticipate that as im-SRT technologies improve, more gene species will become simultaneously detectable in a single experiment. We therefore sought to investigate the extent to which cell type-specific biases in transcriptomics analysis results will be observed with larger gene panels. To evaluate the impact of the size of the skewed gene panel on the extent of cell type or region-specific effects observed, we used full-transcriptome scRNA-seq data collected from sorted peripheral blood mononuclear cells [29] to simulate skewed gene panels of different sizes. Specifically, we simulated gene panels of 50, 100, 500, 1000, and 5000 genes overexpressed in monocytes in addition to a panel including all genes that passed quality filters. We quantified gene panel skew and found that as gene panel size increases, gene panel skew decreases (Additional file 1: Fig. S11). When using library size normalization, cell type-specific differences are observed in the skewed gene panels of all sizes. However, the extent of these differences varies depending on the size of the panel. For the monocyte-specific gene panel, larger differences are observed with the smaller gene panels compared to the larger panels (Fig. 6). Overall, these results indicate that when normalization scaling factors are derived based on gene counts, region- or cell type-specific impacts on

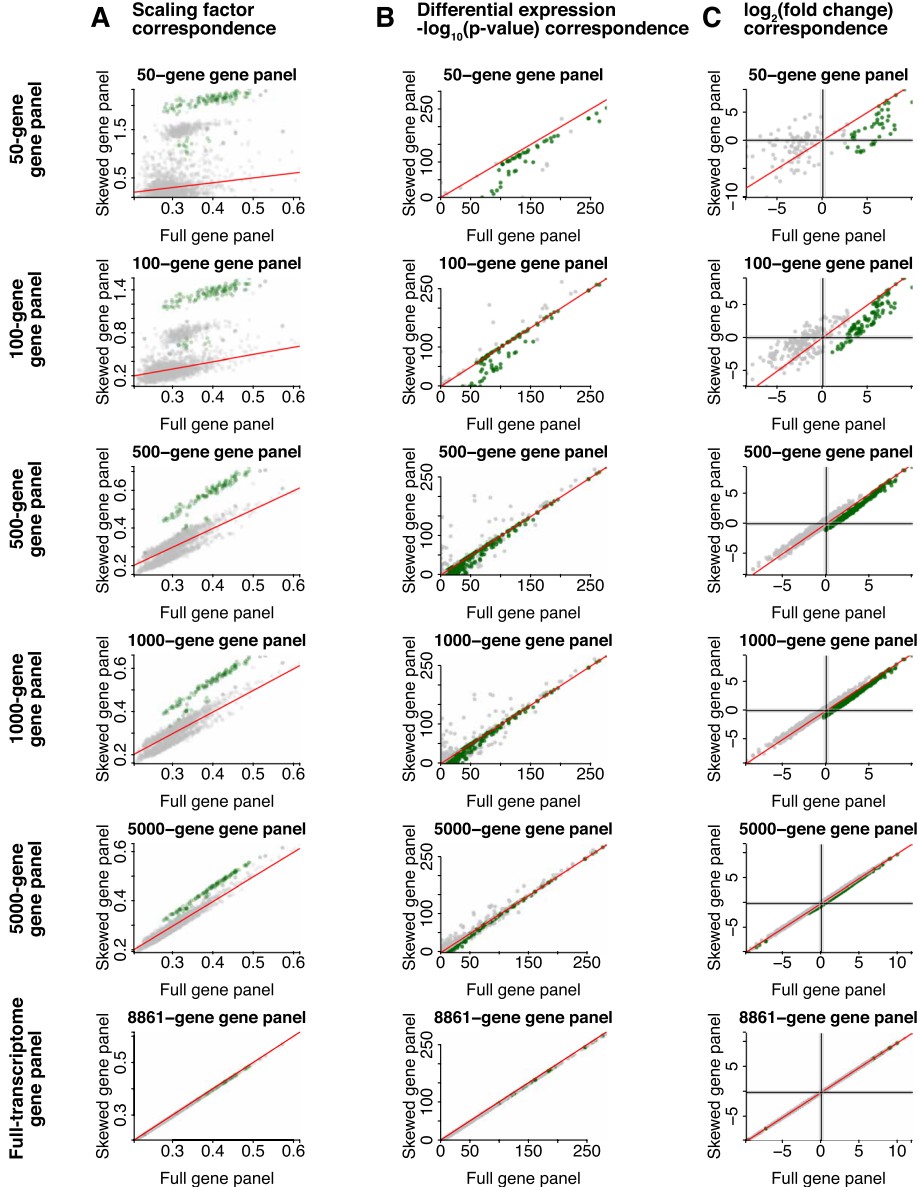

**Fig. 6** Impact of increasing skewed gene panel size on scaling factor, differential expression, and fold change analysis after library size normalization. **A** Scaling factors with monocyte-skewed gene panels of different sizes and full gene panel. Green indicates monocyte cells and grey indicates B cells, T cells, or NK cells. **B** Wilcoxon rank sum test $-\log_{10}(p$-values) adjusted for multiple hypothesis testing for one cell-type-vs-all differential gene expression tests with monocyte skewed gene panels of different sizes and full gene panel. **C** One region-vs-all gene $\log_2$ fold change with monocyte skewed gene panels of different sizes and full gene panel. In **B** and **C**, green indicates monocyte-vs-all tests and grey indicates B cell-, T cell-, or NK cell-vs-all tests

downstream analysis may persist with gene panels as large as 5000 genes, though the extent of these impacts will depend on the normalization method used as well as the degree of region- or cell type-specific gene expression overrepresentation in the gene panel, with larger gene panels less likely to result in downstream region- or cell type-specific biases compared to smaller gene panels.

## Discussion

Here, we investigate the impact of different normalization methods with different gene panels in the analysis of im-SRT data. In particular, we simulated gene panels that over-represent the gene expression of specific tissue regions or cell types. Given im-SRT data with these skewed gene panels, normalization methods that use count scaling factors derived from detected gene counts such as library size, scTransform, DESeq2, or TMM differentially impact normalized gene expression magnitudes of cells in a region- or cell-type-specific manner. We show that these region or cell type-specific effects may reduce the reliability of downstream differential and spatially variably gene expression analysis. We observe that these results are consistent across multiple simulated skewed gene panels across multiple im-SRT data sets spanning several im-SRT technologies and tissues, demonstrating that the results are independent of upstream, technology-specific technical factors.

In general, count normalization methods have been developed to account for variation in detected gene counts across cells due to technical factors, such as variation in RNA capture rate in scRNA-seq [11]. In im-SRT, variation in the proportion of cell volume imaged can result in variation in detected gene counts. For cells that lie at the boundaries of the imaged regions, not all the cell volume will be captured resulting in a smaller gene count that is reflective of technical factors rather than biological ones. Similarly, for large cells that are not fully captured in imaged regions, cell orientation with respect to planes of imaging, can impact proportion of cell volume captured and in turn number of detected gene counts. We show in simulated gene expression data that when detected gene count differences due to partial volume capture compose a large proportion of total gene expression variation, gene expression normalization within individual im-SRT experiments may be needed to allow for reliable downstream analysis (Additional file 1: Fig. S12).

The gene count-based normalization methods evaluated in this work were developed for use with full-transcriptome RNA sequencing data and rely on assumptions that may be violated in targeted transcriptome profiling such as in im-SRT data, particularly with skewed gene panels. For example, library size normalization assumes that there are no genes that are only highly expressed in one subset of samples; that is, all samples have similar total gene expression [9, 10]. This assumption is inherently violated by designing gene panels to include canonical cell type markers for cell types of interest. scTransform models gene counts using a generalized linear model with sequencing depth (i.e. library size) as the explanatory variable to regress out the effect of sequencing depth. In this way, scTransform treats library size as a proxy for technical variables affecting gene count detection, such as RNA capture rate. In im-SRT, this may be confounded if gene count detection is additionally affected by differences in representation of different cell type gene expression profiles within measured genes. As such, we anticipate that scTransform could be modified to consider alternative or additional explanatory variables such as cell volume in the future. Other normalization methods that attempt to account for such compositional differences in gene expression, such as those used in the DESeq2 and TMM, rely on other assumptions. For example, DESeq2 assumes that fewer than half of genes in the data are differentially expressed between samples. Furthermore, DESeq2 assumes that count data is not sparse and can compute cell scaling

factors close to 0 or 1 when used with sparse data, reflective of a higher dropout rate and not technical variation in gene count detection. With scaling factors close to 1, DESeq2 normalization behaves similarly to no normalization. In TMM, one sample is chosen to be a reference, thereby also assuming that most measured genes are not differentially expressed between samples [10]. This is often not the case in im-SRT gene panels as they are often selected by identifying differentially expressed genes from prior scRNAseq data. Although our simulated gene panels were intentionally skewed for demonstration purposes, we emphasize that the approach by which these genes were selected can be used to design gene panels for im-SRT experiments and that choosing genes to focus on tissue regions or cell types of interest in general may result in unintentionally skewed gene panels.

Despite this, count-based normalization can still provide robust insights when gene panels are representative of most of the component cell types in the tissue being assayed. Indeed, several publicly available im-SRT datasets were generated with the intention of broad spatial characterization of tissue spatial architecture and localization of component cell types. These studies in turn use gene panels that are broadly representative of the gene expression of the tissue's component cell types are therefore less susceptible to region- or cell type-specific biases in downstream analysis resulting from gene count-based normalization. Furthermore, as indicated in our results, biases in downstream transcriptomic analyses, particularly for fold change direction, tend to affect gene expression trends of more modest magnitude. Therefore, we expect strong biological signals to remain consistent even when using count-based normalization with skewed gene panels.

Gene count normalization is an important step in the analysis of transcriptomics data as it precedes many downstream analyses. However, many other upstream experimental and computational factors may also impact the reliability of downstream transcriptomic analyses. These factors include molecule detection sensitivity and specificity, which can vary across im-SRT technologies depending on the molecule probe design, signal amplification approach, barcoding strategy used, uniformity of tissue permeability and perfusion, among others [30–33]. Furthermore, cell segmentation continues to be a challenging step in im-SRT data analysis [34]. Accuracy of cell segmentation directly impacts accuracy of molecule-to-cell assignment and, in turn, impacts sensitivity and specificity of cell count detection [35]. Additional investigation evaluating the impact of factors upstream of gene count normalization, including cell segmentation accuracy, as well as molecule detection sensitivity and specificity on in the analysis of im-SRT data is currently being explored [30, 35, 36]. Likewise, additional metrics to quantify the accuracy of cell segmentation will be important in assessing the impact of cell segmentation errors on downstream analysis of imSRT data [37]. Still, independent of these upstream technology-specific technical factors, experimental design choices—namely gene panel composition and count normalization method—present additional sources of variability and can introduce systematic biases affecting downstream analysis (Additional file 2: Mathematical Notes 1, Additional file 1: Fig. S13-14).

As investigators move beyond tissue cell type characterization and towards utilizing im-SRT to answer targeted, hypothesis-driven questions about specific biological processes or cell types, we anticipate that more targeted gene panels will be needed. In fact,

specialized gene panels skewed towards specific cell subpopulations are already commercially available [19]. In these settings, it will be vital to choose gene expression normalization methods that mitigate the effects of such skewed gene panels on downstream transcriptomic analyses.

## Conclusions

Overall, these results demonstrate that choice of gene expression normalization method in combination with choice of gene panel may impact downstream transcriptomic analyses in a way that affects biological interpretation of im-SRT data.

To this end, we provide a series of recommendations to guide the choice of normalization for im-SRT data (Fig. 7). We recommend, as first choice, cell volume normalization to account for such partial cell capture within imaged regions. Cell area may serve as a proxy where volume estimates are not available. Our results further show that these normalization approaches may be preferred when working with skewed gene panels as scaling factors are independent of gene counts and as such gene panel differences do not impact downstream differential expression, fold change, or spatial gene expression variation analyses. Notably and as seen in the publicly available im-SRT datasets used in this work, cell volume data is not always readily available (Fig. 4). However, based on the results in this study, we anticipate that normalization of gene expression magnitudes by cell volume to account for variation in captured cell volume will emerge as an especially important approach to correcting for within-experiment technical variation in a reliable manner. We therefore encourage investigators and commercial providers of im-SRT technologies to make cell volume estimates readily accessible.

Still, the reliability of cell volume as a normalization factor is contingent upon the accuracy of cell segmentation. Segmentation approaches used vary substantially across studies [18, 38, 39] and can result in segmented cell regions that are not representative of cell morphologies [40]. Because cell segmentation remains a challenging problem, with novel experimental and computational approaches emerging rapidly [41, 42], investigators using publicly available data with accessible cell volume or cell area estimates should therefore carefully assess the reliability of cell volume or cell area estimation methods, before using these data for normalization.

In cases where cell volume or cell area data is not readily accessible or reliable, we recommend the use of non-spatially resolved single-cell resolution, full transcriptome profiling approaches such as scRNA-seq data to inform gene panel design. This can be achieved by generating scRNA-seq and im-SRT datasets from the same tissue, with

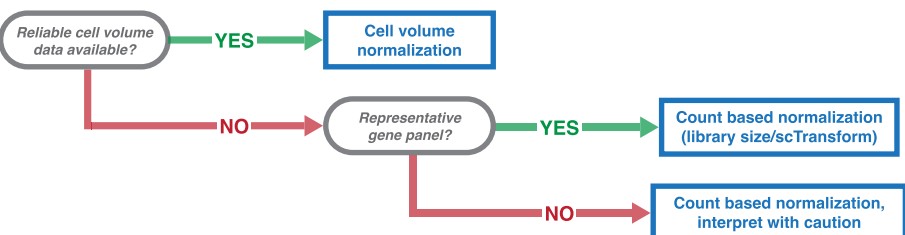

**Fig. 7** Decision tree summarizing recommendations for gene count normalization method selection for im-SRT data

im-SRT panels designed based on differentially expressed genes identified from scRNA-seq data, as is demonstrated in several previous studies [14, 18, 39]. Furthermore, as collective atlasing efforts such as the Human BioMolecular Atlas Program (HuBMAP) [43], the Human Cell Atlas [44], and others continue to generate tissue-specific full-transcriptome gene expression data, we anticipate these resources may help provide a more untargeted view of gene expression across cell types in tissues of interest and enable researchers to design im-SRT gene panels that are representative of most of tissues' component cell types. These resources can also be used to evaluate gene panel skew when designing or using commercially available off-the-shelf gene panels prior to downstream gene expression normalization and transcriptomic analysis. In cases where gene panels are designed to be broadly representative of all cell types within the tissue under investigation, we anticipate that single-cell gene count-based gene expression normalization approaches such as library size or scTransform normalization will be sufficiently reliable for reliable downstream analysis.

Finally, in cases where gene panels are skewed towards a tissue region or cell type of interest, but cell volume or cell area information is unavailable, we still recommend utilizing a gene count-based normalization approach to account for variation in captured cell volume. However, we recommend interpreting results of downstream analyses with caution. As shown in our results, while larger magnitude gene expression effects tend to remain evident with different gene panels, smaller effects are more susceptible to fold change direction changes while still remaining significant on differential expression testing. For this reason, we caution against over-interpreting biological conclusions from effects with small magnitude gene expression changes, and recommend further investigation of candidate effects.

Ultimately, normalization of im-SRT data should enable the removal of the effects of systematic technical variation in detected gene counts. Accomplishing this goal without introducing additional biases will help ensure that normalized gene expression magnitudes reflect underlying biological heterogeneity to enable reliable and reproducible interpretation of im-SRT data in a way that allows investigators to extract meaningful biological insights.

## Methods

### Datasets

Vizgen MERFISH Mouse Brain Receptor Map data was downloaded from the Vizgen website and data from slice 2 replicate 2 was used [20]. Brain anatomical region annotations for each cell were obtained by transferring region annotations from the Allen Brain Atlas Common Coordinate Framework (CCF) using STalign [21, 22]. STalign was applied using the 3D reconstructed Nissl image from the Allen CCF atlas as a source and the MERFISH cell position data as a target. Anatomical regions were combined to obtain coarse anatomical annotations as specified in Additional file 3: Table S1. Cells in the ventricle, fiber tract, dentate gyrus, and habenula regions were used for further analysis.

STARmap PLUS data downloaded from the Broad Single Cell Portal and data from well 11 (well11raw_expression_pd.csv.gz) was used [45]. Brain anatomical region annotations were obtained from previous manually created annotations (well11_spatial.csv.gz) [46]. Anatomical regions were combined to obtain coarse anatomical annotations.

Fiber_tracts_A, Fiber_tracts_C, and Fiber_tracts_D were combined to form the fiber tracts region; DG_A and DG_B formed the dentate gyrus region; VS_A and VS_B formed the ventricle region. Cells in the ventricle, fiber tract, and dentate gyrus regions were used for further analysis.

seqFISH data was downloaded from the Spatial Genomics website: SG_MouseKidney-DataRelease_CxG_section1.csv,

SG_MouseKidneyDataRelease_CellCoordinates_section1.csv [24].

Kidney anatomical regions (cortex, medulla, pelvis) were manually annotated based on tissue architecture. Cells in the cortex, medulla, and pelvis regions were used for further analysis.

CosMx data was downloaded from the NanoString website (LiverDataReleaseSeurat_noTranscripts_newUMAP.rds) as a Seurat object [25]. Gene counts and cell position information were obtained from the "RNA" assay and the "meta.data" slots in the Seurat object. Author annotations were used to identify cells in Zone 1 and Zone 3, which were used for further analysis.

10X Xenium data was downloaded from the 10X Genomics website and replicate 1 was used [26]. Author-provided histological annotations based on hematoxylin and eosin staining defining ductal carcinoma in situ and invasive carcinoma were used to identify tissue regions [40]. Cells within the ductal carcinoma in situ and invasive carcinoma regions were used for further analysis.

Single-cell RNA sequencing data was obtained from the 10X Genomics website [29]. Specifically data from the following datasets was downloaded and combined: human CD14+monocytes, CD19+B-cells, CD8+cytotoxic T-cells, CD4+helper T-cells, CD4+/CD45RO+memory T-cells, CD4+/CD25+Regulatory T-cells, CD8+/CD45RA+Naive Cytotoxic T-cells, CD4+/CD45RA+/CD25- naive T-cells, and CD56+natural killer cells.

### Scaling factors and gene expression normalization

Library size scaling factors were computed as the sum of all gene counts in a cell. DESeq2 scaling factors were computed using the estimateSizeFactorsForMatrix function from the DESeq2 package (version 1.38.0), with method=poscounts. TMM scaling factors were computed as the inverse of the output of the calcNormFactors function from the edgeR package (version 3.40.0), method=TMMwsp. For the MERFISH data, cell volume scaling factors were included in the downloaded data. For seqFISH, CosMx, and 10X Xenium data, cell area was included in the downloaded data. To obtain normalized gene expression, gene counts for each gene in each cell were divided by that cell's scaling factor. To normalize gene counts using scTransform, we used the vst function in the sctransform package to obtain the Pearson residuals, which were taken to be the normalized gene counts after setting negative counts to 0.

### Differential gene expression testing and fold change analysis

To test for differential gene expression, for each gene in each region, a one-sided Wilcoxon rank sum test was performed between cells in that region and all other cells in the data. Significantly differentially expressed genes were identified using a *p*-value cutoff of 0.05 after

a Benjamini–Hochberg multiple hypothesis correction. For each gene in each region, $\log_2$ fold change (*LFC*) was computed as follows:

$$LFC_{A,g} = log_2 \frac{\frac{1}{N_A} \sum_i^{N_A} x_{g,i}}{\frac{1}{N_{A'}} \sum_j^{N_{A'}} x_{g,j}}$$

Where $LFC_{A,g}$ is the $log_2$ fold change for gene $g$ between cells in region $A$ and all other cells in the data, $x_{g,i}$ is the normalized gene expression of gene $g$ in cell $i$, $N_A$ is the number of cells in region $A$, and $N_{A'}$ is the number of cells in data in regions other than in region $A$.

### Spatially variable gene expression analysis

To identify spatially variable genes after gene expression normalization, we first rasterized the normalized gene expression data using SEraster for computational tractability, using a bin size of 50 μm, and averaging counts within each bin [28]. Rasterized gene expression was then $\log_{10}$ transformed with a pseudocount of 1.

We then used nnSVG package to identify spatially variable genes [27].

### Simulation of skewed gene panels in im-SRT data

Differential gene expression testing was performed on the normalized counts using the original, full gene panel for each normalization method as well as no normalization. After differential gene expression testing, significantly differentially expressed genes for a region were identified as those genes with a *p*-value < 0.05 after multiple hypothesis correction and an absolute $\log_2$ fold change > 0.25 under any normalization procedure. The skewed gene panel was simulated by randomly selecting genes from those identified as significantly differentially expressed. The random gene panel was simulated by randomly selecting the same number of genes as in the skewed gene panel but from all the genes in the data. Gene panels of size 100, 100, 60, 200, and 80 genes were simulated for the MERFISH, STARmap PLUS, seqFISH, CosMx, and 10X Xenium datasets, respectively.

### Simulation of skewed gene panels in scRNA-seq data

Differential gene expression testing was performed on the normalized counts using the original, full gene panel for each normalization method as well as no normalization. Genes were ranked according to their multiple hypothesis corrected *p*-value. Skewed gene panels were selected by choosing the most significantly differentially expressed genes.

### Quantifying gene panel skew

Gene panel skew was computed as the Kullback–Leibler divergence between the observed gene counts in each region and that regions' expected gene counts given cell proportions within each region.

$$Skew = \sum_i^{N_R} P_i \ln\left(\frac{P_i}{Q_i}\right)$$

Where $P_i$ is the proportion of total detected gene counts in cell subpopulation $i$, $Q_i$ is the proportion of cells in cell subpopulation $i$, and $N_R$ is the total number of cell subpopulations. Cell subpopulations can be any grouping of cells in the data, such as tissue anatomical region or cell type. Here, we are assuming that cells in all cell subpopulations have comparable total gene expression and that systematic differences in detected counts across cells in different cell subpopulations are due to gene panel skew.

**Computing differential expression and spatially variable gene identification error rates**

Differential expression testing false positive and false negative rates were computed as follows:

$$FPR = \frac{FP}{FP + TN}, FNR = \frac{FN}{FN + TP}$$

Where *FP* is the number of significant tests with the skewed gene panel where the corresponding test with the full gene panel is not significant, *FN* is the number of non-significant tests with the skewed gene panel where the corresponding test with the full gene panel is significant, and *TP* and *TN* are the number of tests that are significant and non-significant with the both the skewed and full gene panels, respectively. Significance threshold used was $p < 0.05$ after Benjamini–Hochberg multiple hypothesis correction.

**Computing fold change switch rates**

Fold change switched positive and switched negative rates were computed as follows:

$$SPR = \frac{SP}{SP + TN}, FNR = \frac{SN}{SN + TP}$$

Where *SP* is the number of positive $\log_2$ fold changes with the skewed gene panel where the corresponding comparison with the full gene panel is negative, *SN* is the number of negative $\log_2$ fold changes with the skewed gene panel where the corresponding comparison with the full gene panel is positive, and *TP* and *TN* are the number of comparisons that are positive and negative with the both the skewed and fill gene panels, respectively.

**Simulating partial cell volume and gene expression capture**

Synthetic gene expression profiles for 500 genes in 2000 cells were simulated using the package Splatter using parameters estimated from the MERFISH mouse brain dataset using the splatEstimate function [47].

The 2000 simulated cells were divided into two subpopulations, A and B. Cell subpopulation A was simulated to be spatially distributed along the *Z*-axis following a normal distribution centered at a mean of 4.839 μm and a standard deviation of 1.5 μm. Cell subpopulation B was simulated to be spatially distributed along the *Z*-axis following a normal distribution centered at a mean of 12 μm and a standard deviation of 1.5 μm.

Ground truth cell volumes were simulated to be normally distributed with a mean of 1000 μm$^3$, based on an average cell radius on the order of 10 μm, and a standard deviation of 2000 μm$^3$.

We simulated cell volume capture from 7 *Z* planes of imaging spaced 1.5 μm apart. We further assumed an imaging depth of field of 0.678 μm, based on imaging at 550 nm,

with a $\times$ 60 1.4NA objective with immersion oil. Based on a cell's Z position, it was assigned to one of the 5 following groups, and proportion of cell volume captured was computed accordingly:

Group 1: cells whose center is above the top imaged Z plane, with a portion of volume captured. Here, proportion of cell volume captured, $P_v$, is computed as:

$$P_v = \frac{h_c^2(3r_c - h_c)}{4r_c^3}$$

Where $r_c$ is the cell radius, $h_c$ is the extent of the cell captured in the Z dimension based on its simulated Z position, with $h_c = r_c - (z_c - z_{top})$, where $z_c$ is the simulated Z position of the cell center, and $z_{top}$ is the Z position of the top imaged Z plane.

Group 2: cells whose center is below the top imaged Z plane, with a portion of volume captured. Here, proportion of cell volume captured, $P_v$, is computed as:

$$P_v = 1 - \frac{h_c^2(3r_c - h_c)}{4r_c^3}$$

Where $r_c$ is the cell radius, $h_c$ is the extent of the cell not captured in the Z dimension based on its simulated Z position, with $h_c = r_c + z_c - z_{top}$, where $z_c$ is the simulated Z position of the cell center, and $z_{top}$ is the Z position of the top imaged Z plane $+ 0.5$*depth of field.

Group 3: cells fully captured within imaged planes. Here, proportion of cell volume captured $= 1$.

Group 4: cells whose center is above the bottom imaged Z plane, with a portion of volume captured. Here, proportion of cell volume captured, $P_v$, is computed as:

$$P_v = 1 - \frac{h_c^2(3r_c - h_c)}{4r_c^3}$$

Where $r_c$ is the cell radius, $h_c$ is the extent of the cell not captured in the Z dimension based on its simulated Z position, with $h_c = r_c - z_c$, where $z_c$ is the simulated Z position of the cell center, and $z_{top}$ is the Z position of the top imaged Z plane.

Group 5: cells whose center is below the bottom imaged Z plane, with a portion of volume captured. Here, proportion of cell volume captured, $P_v$, is computed as:

$$P_v = \frac{h_c^2(3r_c - h_c)}{4r_c^3}$$

Where $r_c$ is the cell radius, $h_c$ is the extent of the cell captured in the Z dimension based on its simulated Z position, with $h_c = r_c + z_c$, where $z_c$ is the simulated Z position of the cell center, and $z_{top}$ is the Z position of the top imaged Z plane.

Captured gene counts were computed by multiplying each cell's simulated gene expression profile by its proportion of cell volume captured.

Gene expression profiles were visualized in principal component space by reducing to 50 dimensions, after $\log_{10}$ transforming counts with a pseudocount of 1, and centering and scaling each gene's transformed counts to have a mean of 0 and standard deviation of 1, using the reduceDimenions function in the veloviz package.

Normalization by library size, DESeq2, and cell volume and differential expression testing was performed as described above.

## Review history

The review history is available as Additional file 4.

## Peer review information

## Supplementary Information

---

Additional file 1: Figures S1-14

Additional file 2: Mathematical Note 1

Additional file 3: Table S1: Specification of course mouse brain anatomical region annotations

Additional file 4: Review history

---

**Authors' contributions**

LA and KC conceptualized the study. LA and JF designed the methodology. LA performed analysis. KC, MA, and GA contributed to analysis under the guidance of LA. LA and JF wrote the manuscript. All authors contributed to the reviewing and editing of the manuscript.

**Funding**

Research reported in this publication was supported by the National Institute of General Medical Sciences of the National Institutes of Health under Award Number R35-GM142889 and the National Science Foundation under Grant No. 2047611.
National Institute of General Medical Sciences,R35-GM142889,Jean Fan,Division of Biological Infrastructure,2047611,Jean Fan

**Availability of data and materials**

Code to reproduce the analyses and results of this study is available on Github at https://github.com/LylaAtta123/normalization-analyses [48].
MERFISH mouse brain: Vizgen MERFISH Mouse Brain Receptor Map, slice 2 replicate 2 [20].
STARmap PLUS mouse brain: Broad Single Cell Portal, Spatial Atlas of Molecular Cell Types and AAV Accessibility across the Whole Mouse Brain [45].
seqFISH mouse kidney: Spatial Genomics, Sample mouse kidney, Sect. 1:
SG_MouseKidneyDataRelease_CxG_section1.csv,
SG_MouseKidneyDataRelease_CellCoordinates_section1.csv [24].
CosMx liver data: NanoString, CosMx SMI Human Liver FFPE Dataset—Seurat Object (no transcripts): LiverDataReleaseSeurat_noTranscripts_newUMAP.rds [25].
10X Xenium: 10X Genomics, Xenium FFPE Human Breast with Custom Add-on Panel, Sample 1, replicate 1[26].
Peripheral blood mononuclear cell single-cell RNA sequencing: 10X Genomics, human CD14 + monocytes, CD19 + B-cells, CD8 + cytotoxic T-cells, CD4 + helper T-cells, CD4 +/CD45RO + memory T-cells, CD4 +/CD25 + Regulatory T-cells, CD8 +/CD45RA + Naive Cytotoxic T-cells, CD4 +/CD45RA +/CD25- naive T-cells, and CD56 + natural killer cells, [29].

## Declarations

**Ethics approval and consent to participate**
Not applicable.

**Consent for publication**
Not applicable.

**Competing interests**
The authors declare that they have no competing interests.

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

## 