## [Additional file 4: Review history · Genome Biology]

Review History

First round of review

Reviewer 1

Are you able to assess all statistics in the manuscript, including the appropriateness of statistical tests used? No, I do not feel adequately qualified to assess the statistics.

Comments to author:

Atta et al. evaluated the performance of 4 normalization strategies on image-based ST under different settings. The tested normalization strategies include library size, DEseq2, TMM, and cell area. The tested image-based spatial transcriptome (ST) data include MERFISH and STARmapPLUS. Panel settings were simulated by selecting gene subset of original selected genes. Overall, we think this benchmarking study is necessary for its initial position in a ST analytical pipeline. We have the following suggestions to improve the work.

The MS tries to convey messages to general imaging-based ST, but the included ST data types are rather limited (only 2 types), far less than currently available data types of at least 10 types (<https://doi.org/10.1038/s41592-022-01409-2>). We think this work can benefit from including more im-ST data types for example seqFISH, seqFISH+, osmFISH, etc. Especially those 3D types such as STARmap and EASIFISH.

Similarly, conclusions in this study were drawn from only brain cannot generate to other organs. We suggest at least 4 organ types should be considered.

The included normalization strategies are also limited, is there no other methods for count normalization for single cell or ST data?

The author correctly stated that im-ST bulk the RNA molecules spots within cell regions to achieve single-cell resolution. But the cell-level expressions are fundamentally determined by cell segmentation algorithms and accuracy. It can also influence the cell size which was considered as a normalization factor by the authors. How does the cell segmentation algorithms and accuracy influence the performances of normalization strategies? Do different degrees of segmentation inaccuracies have large effect on the conclusions?

The authors are also encouraged to consider more upstream factors that influence the gene expression quantification, for example the applied molecule spots detection methods and its parameters/accuracy.

The MS majorly used scatter plots to evaluate different normalization strategies, but lacks the use of scalar value metrics for a more straightforward message. We suggest using the performance of downstream applications to improve this point. For example, the performance of spatial domain (<https://doi.org/10.1038/s41467-022-29439-6>), cell typing (<https://doi.org/10.1038/s41592-021-01264-7>), or spatially variable gene (<https://academic.oup.com/nar/article/51/20/e103/7301277>) etc.

The authors mentioned different gene panel selection methods, these panel selections should be considered, instead just using random selection or region specific selection.

The major conclusions of this manuscript are (1) region specific gene panel negatively impact downstream analysis, and (2) more representative gene panel can mitigate some negative impact. These are within expectation. The manuscript would benefit from using more comprehensive datasets and metrics to derive a more useful guidelines for users or propose a computational solution or recommendations to address some limitations of current normalization methods. Otherwise we cannot see direct contributions of this work.

We think other methods evaluation papers on similar topics such as <https://doi.org/10.1038/s41467-022-35740-1> and <https://doi.org/10.1038/s41467-023-37168-7> should be helpful to improve this work.

Reviewer 2

Are you able to assess all statistics in the manuscript, including the appropriateness of statistical tests used? Yes, and I have assessed the statistics in my report.

Comments to author:

In this manuscript, Atta et al. study the effects of various normalization methods applied to imaging-based spatial transcriptomics data. These techniques are currently targeting only a "small" set of selected genes and the authors show that if these genes are biased in the proportion of markers for distinct cell types, common ("library size", DESeq2 and edgeR) normalization scaling factors are consequently biased which impacts downstream differential analysis of gene expression visualization. The authors also make very clear that cell volume-based normalization is not affected by the composition of the gene panel (since it is based on the geometry of the cell instead of its observed expression data). Using simulations, they show that this effect is particularly pronounced on small gene panels, which corresponds to the current state of the technology.

The paper is well written and easy to follow. I do not have technical issues with the data presented. I however feel that while the authors make very clear that volume-based normalization is not affected by gene panel composition, unlike other normalization methods based on expression data, they do not address whether volume-based normalization is practically better for downstream analyses such as differential expression analysis. I therefore question whether the mention of volume-based normalization and its inclusion in the figures is really warranted without a comparison of volume-based normalization with other normalization methods on downstream analysis of actual data.

Minor points :

- In scatter plots, it would be nice to a visual legend indicating that purple/green dots come from the region enriched in differentially-expressed genes for that region
- Line 388 : typo ("NA' and is" instead of "and NA' is")
- Simulation of skewed gene panels in im-SRT data : it could be interesting to generate random panels with various proportions of DEGs to see the impact of this proportion differential expression

Gene count normalization in single-cell imaging-based spatially resolved transcriptomics: Point-by-point Reviewer Response

Overview:

We sincerely thank the editor and the reviewers for their insightful and constructive feedback in helping us improve this manuscript. We have now revised the manuscript to address all the points raised by the reviewers, organized herein as a point-by-point response. Throughout this point-by-point response, reviewer comments are shown in **blue**, with our responses in **green**, and changes to the manuscript in **black**.

Reviewer #1:

Atta et al. evaluated the performance of 4 normalization strategies on image-based ST under different settings. The tested normalization strategies include library size, DEseq2, TMM, and cell area. The tested image-based spatial transcriptome (ST) data include MERFISH and STARmapPLUS. Panel settings were simulated by selecting gene subset of original selected genes. Overall, we think this benchmarking study is necessary for its initial position in a ST analytical pipeline. We have the following suggestions to improve the work.

1. The MS tries to convey messages to general imaging-based ST, but the included ST data types are rather limited (only 2 types), far less than currently available data types of at least 10 types (<https://doi.org/10.1038/s41592-022-01409-2>). We think this work can benefit from including more im-ST data types for example seqFISH, seqFISH+, osmFISH, etc. Especially those 3D types such as STARmap and EASIFISH.

We thank the reviewer for the recommendation to investigate additional data types. As recommended, we investigate data from 3 additional imaging-based spatially resolved transcriptomics (imSRT) technologies: 10X Genomics Xenium data from human breast cancer, seqFISH from mouse kidney, and CosMx from human liver [1], [2], [3]. As with the MERFISH and STARmapPLUS analyses, we design region-specific gene panels to investigate the effects of different normalization procedures with gene panels that overrepresent gene expression of certain tissue regions. Specifically, for the 10X Genomics Xenium data, we simulate gene panels specific to ductal carcinoma in situ and invasive carcinoma. For the seqFISH data, we simulate gene panels specific to the kidney cortex, medulla, and pelvis. For the CosMx data, we simulate gene panels specific to Zone 1 and Zone 3 of the liver. For each dataset, we also simulate a gene panel with randomly sampled genes, to investigate the impact of normalization procedures with similarly-sized but representative gene panels.

With these additional datasets, we perform similar analyses to those we performed with the MERFISH and STARmapPLUS data. Overall, we observe similar results similar to those we observe in with MERFISH and STARmapPLUS data.

In particular, we investigate normalization using count-based procedures (library size, scTransform, DESeq2, and TMM) as well as cell area or volume as was available and no normalization. Looking at scaling factors, we find that with count-based normalization methods, scaling factors are differentially affected in a region-specific manner. Specifically, cells in the region to which the gene panel is skewed systematically had higher scaling factors than cells in other regions. We observe this effect with each of the additional data sets. We note that scTransform does not use cell specific scaling factors, and so this comparison was omitted.

We then normalized gene expression counts using these scaling factors and evaluated the correlation between normalized gene expression magnitudes when using the skewed gene panels and corresponding full gene panels with the same normalization. Similarly, to previous results we find that count-based normalizations including library size, scTransform, and TMM generally have lower and more variable correlations compared to DESeq2. Correlations of normalized gene expression magnitudes after cell area or volume normalization or no normalization are 1 since scaling factors are independent of gene panel.

Next, we investigated the effect on differential gene expression testing and gene expression fold change analysis. As before, we compared the p-value of a Wilcoxon rank sum test of differential expression as well as the gene fold change between cells in each brain region and all other regions based on normalized gene expression magnitudes achieved with the skewed gene panel versus the full gene panel for each normalization method. When looking at p-values after normalization with count-based methods, we find that genes specific to the region to which the gene panel is skewed tend to have less significant p-values with the skewed gene panel compared to the full gene panel. This effect is observed across skewed gene panels and across data sets and is especially evident with library size and scTransform normalization. When looking at gene expression fold change, we find that with count-based normalization methods, genes specific to the region to which the gene panel is skewed tend to have lower log₂ fold changes when using the skewed gene panel compared to the full gene panel, with some genes having negative log₂ fold changes in the skewed gene panel compared to positive log₂ fold changes in the full gene panel. Again, these effects are observed across skewed gene panels and across datasets. We quantify these results by computing the error rates in p-values and log₂ fold changes between the skewed and corresponding normalized gene panels. This is discussed in detail in the response to reviewer comment #8.

As before, we also investigated the effect of different gene expression procedures when using gene panels of the same size as the region-specific gene panels above, but with genes selected at random so as to be more representative of the overall tissue gene expression. We find that the region-specific effects observed above are mitigated when gene panels are more representative of the overall gene expression of tissue regions assayed.

Finally, we investigated the impact on spatially variable gene analysis which we will discuss in detail below in the response to comment #6.

We have incorporated these additional datasets and their associated results into the revised manuscript and provided below for the reviewer's convenience:

Revised text (Results lines 265-289, Figures 3-4, Supplementary Figures 5-8):

To further investigate if these observations extend to im-SRT data acquired by other technologies and from other tissues, we performed the same analyses with a number of other im-SRT datasets. We evaluated the effects of different normalization approaches with simulated skewed gene panels in STARmap PLUS data from mouse brain, seqFISH data from mouse kidney, CosMx data from human liver, and 10X Xenium from human breast cancer [26] (Supp Fig 5-8). Skewed gene panels were simulated by sampling differentially expressed genes for regions identified manually based on tissue architecture (seqFISH mouse kidney), from author provided pathology annotations (STARmap PLUS mouse brain, 10X Xenium human breast cancer), or from author provided annotations derived from spatial-cell type clustering (CosMx human liver). We observe similar region-specific effects on scaling factors, differential gene expression p-values, and gene \log_2 fold changes across im-SRT technologies and tissues. We quantify differential expression testing error rates and fold change switch rates (Figure 4). While error rates vary by dataset and simulated gene panel, we generally observe higher differential expression error rates with library size and scTransform normalization with skewed gene panels, reaching up to 17% with some simulated gene panels. Similarly, we generally observe higher fold change switch rates with library size and scTransform normalization reaching up to 32% with some simulated gene panels. We note that we observe higher rates of fold change switching at more modest fold changes compared to at higher fold changes where fold change directions tends to remain consistent between skewed gene panels and their corresponding full gene panel.

Overall, these results show that the region-specific effects of gene count-based normalization in im-SRT with skewed gene panels may result in false positives and negatives in differential gene expression testing, as well as fold changes with opposite directions. Further, these results suggest that region-specific differences in transcriptomics analysis results using skewed gene panels also generalize to im-SRT data from other imaging-based spatial transcriptomics technologies.

Figure 1. im-SRT normalization evaluation workflow.

- (1) 5 publicly available im-SRT datasets representing a variety of tissues and im-SRT technologies. Cells colored by tissue region annotations.
- (2) im-SRT datasets with skewed gene panels were simulated by sampling genes overrepresenting different tissue regions.
- (3) Up to 5 gene count normalization approaches were applied to each im-SRT dataset with the simulated skewed gene panels as well as with the original full gene panel.
- (4) Downstream analysis of spatial gene expression data including differential gene expression, gene expression fold change, and spatially variable gene analysis were performed.
- (5) Ideally, biological conclusions would be robust to gene panel choice. Robustness of downstream analysis results after each normalization approach was investigated by comparing results for each im-SRT dataset with each skewed gene panel compared to the corresponding full gene panel.

Figure 3. Impact of region-skewed gene panel on differential expression analysis and fold change evaluation

A. Wilcoxon rank sum test $-\log_{10}(\text{p-values})$ adjusted for multiple hypothesis testing for one region-vs-all differential gene expression test with normalized gene expression from the full gene panel versus the ventricle-skewed gene panel. Red line indicates $x=y$. Inset shows smaller $-\log_{10}(\text{adjusted p-values})$. Purple indicates ventricle-vs-all tests, grey indicates other region-vs-all tests.

B. Library size normalized gene expression magnitudes in cells in region of interest vs cells in other regions with the full gene panel and the ventricle skewed gene panel for the false positive (*Lgr4*, top) and false negative (*Ryk*, bottom) genes highlighted in A.

C. Differential gene expression false positive (top) and false negative (bottom) rates after each normalization method, taking differential gene expression results with the full gene panel to be ground truth.

D. One region-vs-all gene fold change with normalized gene expression from the full gene panel and the ventricle gene panel. Black lines indicate 0 log₂ fold change and grey lines indicate 0.5 log₂ fold change. Purple indicates ventricle-vs-all comparisons, grey indicates other region-vs-all comparisons.

E. Library size normalized gene expression magnitudes in cells in region of interest vs cells in other regions with the full gene panel and the ventricle skewed gene panel for the fold change switched positive (*Htr2c*, top) and switched negative (*Gpr75*, bottom) genes highlighted in D.

F. Gene expression fold change switched positive (top) and switched negative (bottom) rates after each normalization method, taking fold change results with the full gene panel to be ground truth.

Figure 4. Impact of normalization and region-skewed and non-skewed gene panels on differential expression analysis and fold change evaluation for multiple simulated gene panels in 5 analyzed im-SRT datasets.

- A.** Differential gene expression false positive rates after each normalization method with each simulated gene panel.
- B.** Differential gene expression false negative rates after each normalization method with each simulated gene panel.
- C.** Gene expression fold change switched positive rates after each normalization method with each simulated gene panel.
- D.** Gene expression fold change switched negative rates after each normalization method with each simulated gene panel.

Supplementary Figure 5. Impact of gene count normalization methods with region-skewed gene panels in STARmap PLUS mouse brain im-SRT data

- A.** For a coronal section of the mouse brain assayed by STARmap, heatmaps showing scaled mean gene expression of the 100 genes in 100-gene gene panel skewed towards ventricle (left), fiber tract (middle), and dentate gyrus (right) brain regions for the three analyzed brain regions.
- B.** Gene panel skew of all simulated and original full STARmap PLUS gene panels.
- C.** Scatterplots of $\log_{10}(\text{scaling factors})$ with library size normalization for each cell for ventricle (left), fiber tract (middle), and dentate gyrus (right) region-skewed gene panels versus full gene panel. Cells in the ventricle, fiber tract, and dentate gyrus regions are shown in purple, blue, and red, respectively. Cells in other regions are shown in grey. Red line indicates $x=y$.
- D.** Root mean squared error in scaling factors after each normalization method between the different region-skewed gene panels and the full gene panel.
- E.** Wilcoxon rank sum test $-\log_{10}(\text{p-values})$ adjusted for multiple hypothesis testing for one region-vs-all differential gene expression test with library size normalized gene expression from the ventricle (left), fiber tract (middle), and dentate gyrus (right) region-skewed gene panels versus the full gene panel. Purple, blue, and red indicates ventricle-, fiber tract- and dentate gyrus-vs-all tests, grey indicates other region-vs-all tests. Red line indicates $x=y$.
- F.** Differential gene expression false positive (left) and false negative (right) rates after each normalization method, taking differential gene expression results with the full gene panel to be ground truth, for each region-skewed gene panel.
- G.** One region-vs-all gene fold change with library size normalized gene expression from the ventricle (left), fiber tract (middle), and dentate gyrus (right) region-skewed gene panels versus the full gene panel. Purple, blue, and red indicates ventricle-, fiber tract- and dentate gyrus-vs-all comparisons, grey indicates other region-vs-all comparisons. Red line indicates $x=y$.
Black lines indicate $0 \log_2$ fold change and grey lines indicate $0.5 \log_2$ fold change.
- H.** Gene expression fold change switched positive (left) and switched negative (right) rates after each normalization method, taking differential gene expression results with the full gene panel to be ground truth, for each region-skewed gene panel.

Supplementary Figure 6. Impact of gene count normalization methods with region-skewed gene panels in seqFISH mouse kidney im-SRT data

A. For a section of the mouse kidney assayed by seqFISH, heatmaps showing scaled mean gene expression of the 60 genes in 60-gene gene panel skewed towards cortex (left), medulla (middle), and pelvis (right) kidney regions for the three analyzed kidney regions.

B. Gene panel skew of all simulated and original full seqFISH gene panels.

- C.** Scatterplots of \log_{10} (scaling factors) with library size normalization for each cell for cortex (left), medulla (middle), and pelvis (right) region-skewed gene panels versus full gene panel. Cells in the cortex, medulla, and pelvis regions are shown in red, green, and blue, respectively. Cells in other regions are shown in grey. Red line indicates $x=y$.
- D.** Root mean squared error in scaling factors after each normalization method between the different region-skewed gene panels and the full gene panel.
- E.** Wilcoxon rank sum test $-\log_{10}$ (p-values) adjusted for multiple hypothesis testing for one region-vs-all differential gene expression test with library size normalized gene expression from the cortex (left), medulla (middle), and pelvis (right) region-skewed gene panels versus the full gene panel. Red, green, and blue indicates cortex-, medulla- and pelvis-vs-all tests, grey indicates other region-vs-all tests. Red line indicates $x=y$.
- F.** Differential gene expression false positive (left) and false negative (right) rates after each normalization method, taking differential gene expression results with the full gene panel to be ground truth, for each region-skewed gene panel.
- G.** One region-vs-all gene fold change with library size normalized gene expression from the cortex (left), medulla (middle), and pelvis (right) region-skewed gene panels versus the full gene panel. Red, green, and blue indicates cortex-, medulla- and pelvis-vs-all comparisons, grey indicates other region-vs-all comparisons. Red line indicates $x=y$. Black lines indicate $0 \log_2$ fold change and grey lines indicate $0.5 \log_2$ fold change.
- H.** Gene expression fold change switched positive (left) and switched negative (right) rates after each normalization method, taking differential gene expression results with the full gene panel to be ground truth, for each region-skewed gene panel.

Supplementary Figure 7. Impact of gene count normalization methods with region-skewed gene panels in CosMx human liver im-SRT data

A. For a section of human liver assayed by CosMx, heatmaps showing mean gene expression of the 200 genes in 200-gene gene panel skewed towards Zone 1 (left) and Zone 3 (right) regions for the two analyzed liver regions.

B. Gene panel skew of all simulated and original full CosMx gene panels.

C. Scatterplots of $\log_{10}(\text{scaling factors})$ with library size normalization for each cell for Zone 1 (left) and Zone 3 (right) region-skewed gene panels versus full gene panel. Cells in Zone 1 and Zone 3 are shown in red and cyan, respectively. Cells in other regions are shown in grey. Red line indicates $x=y$.

D. Root mean squared error in scaling factors after each normalization method between the different region-skewed gene panels and the full gene panel.

E. Wilcoxon rank sum test $-\log_{10}(\text{p-values})$ adjusted for multiple hypothesis testing for one region-vs-all differential gene expression test with library size normalized gene expression from the Zone 1 (left) and Zone 3 (right) region-skewed gene panels versus the full gene panel. Red and cyan indicates Zone 1 and Zone 3-vs-all tests, grey indicates other region-vs-all tests. Red line indicates $x=y$.

F. Differential gene expression false positive (left) and false negative (right) rates after each normalization method, taking differential gene expression results with the full gene panel to be ground truth, for each region-skewed gene panel.

G. One region-vs-all gene fold change with library size normalized gene expression from the Zone 1 (left) and Zone 3 (right) region-skewed gene panels versus the full gene panel. Red and cyan indicates Zone 1 and Zone 3-vs-all comparisons, grey indicates other region-vs-all comparisons. Red line indicates $x=y$. Black lines indicate $0 \log_2$ fold change and grey lines indicate $0.5 \log_2$ fold change.

H. Gene expression fold change switched positive (left) and switched negative (right) rates after each normalization method, taking differential gene expression results with the full gene panel to be ground truth, for each region-skewed gene panel.

Supplementary Figure 8. Impact of gene count normalization methods with region-skewed gene panels in 10X Xenium human breast cancer im-SRT data

A. For a section of human breast cancer assayed by 10X Xenium, heatmaps showing mean gene expression of the 80 genes in 80-gene gene panel skewed towards invasive carcinoma (left) and ductal carcinoma *in situ* (DCIS, right) regions for the two analyzed regions.

B. Gene panel skew of all simulated and original full 10X Xenium gene panels.

C. Scatterplots of $\log_{10}(\text{scaling factors})$ with library size normalization for each cell for the invasive carcinoma (left) and DCIS (right) region-skewed gene panels versus full gene panel. Cells in the invasive carcinoma and DCIS regions are shown in cyan and red, respectively. Cells in other regions are shown in grey. Red line indicates $x=y$.

D. Root mean squared error in scaling factors after each normalization method between the different region-skewed gene panels and the full gene panel.

E. Wilcoxon rank sum test $-\log_{10}(\text{p-values})$ adjusted for multiple hypothesis testing for one region-vs-all differential gene expression test with library size normalized gene expression from the invasive carcinoma (left) and DCIS (right) region-skewed gene panels versus the full gene panel. Cyan and red indicates invasive carcinoma and DCIS-vs-all tests, grey indicates other region-vs-all tests. Red line indicates $x=y$.

F. Differential gene expression false positive (left) and false negative (right) rates after each normalization method, taking differential gene expression results with the full gene panel to be ground truth, for each region-skewed gene panel.

G. One region-vs-all gene fold change with library size normalized gene expression from the invasive carcinoma (left) and DCIS (right) region-skewed gene panels versus the full gene panel. Cyan and red indicates invasive carcinoma and DCIS-vs-all comparisons, grey indicates other region-vs-all comparisons. Red line indicates $x=y$. Black lines indicate 0 \log_2 fold change and grey lines indicate 0.5 \log_2 fold change.

H. Gene expression fold change switched positive (left) and switched negative (right) rates after each normalization method, taking differential gene expression results with the full gene panel to be ground truth, for each region-skewed gene panel.

The generalization of our results with additional data types from different imSRT technologies provides further support for our recommendation for caution with smaller and less representative imSRT panels, when using count-based normalization approaches. Based on these new results, we expect to see region- or celltype-specific biases in gene expression magnitude normalization using count-based gene expression normalization with gene panels that overrepresent gene expression of a specific tissue region or celltype, irrespective of the imaging-based spatially resolved transcriptomics technology used.

As such, we anticipate such trends to be consistent across tissues and imSRT technologies where targeted gene panels are limited to a few hundred genes. We have elaborated on this further in the discussion, also reproduced below for convenience:

Revised text (Discussion lines 392-448):

Here, we investigate the impact of different normalization methods with different gene panels in the analysis of im-SRT data. In particular, we simulated gene panels that overrepresent the gene expression of specific tissue regions or cell types. Given im-SRT data with these skewed gene panels, normalization methods that use count scaling factors derived from detected gene counts such as library size, scTransform, DESeq2, or TMM differentially impact normalized gene expression magnitudes of cells in a region- or cell-type-specific manner. We show that these region or cell type-specific effects may reduce the reliability of downstream differential and spatially variably gene expression analysis. We observe that these results are consistent across multiple simulated skewed gene panels across multiple im-SRT data sets spanning several im-SRT technologies and tissues. Overall, these results demonstrate that choice of gene expression normalization method in combination with choice of gene panel may impact downstream transcriptomic analyses in a way that affects biological interpretation of im-SRT data.

The gene count-based normalization methods evaluated in this work were developed for use with full-transcriptome RNA sequencing data and rely on assumptions that may be violated in targeted transcriptome profiling such as in im-SRT data, particularly with skewed gene

panels. For example, library size normalization assumes that there are no genes that are only highly expressed in one subset of samples; that is, all samples have similar total gene expression. This assumption is inherently violated by designing gene panels to include canonical cell type markers for cell types of interest. scTransform models gene counts using a generalized linear model with sequencing depth (i.e. library size) as the explanatory variable to regress out the effect of sequencing depth. In this way, scTransform treats library size as a proxy for technical variables affecting gene count detection, such as RNA capture rate. In im-SRT, this may be confounded if gene count detection is additionally affected by differences in representation of different cell type gene expression profiles within measured genes. As such, we anticipate that scTransform could be modified to consider alternative or additional explanatory variables such as cell volume in the future. Other normalization methods that attempt to account for such compositional differences in gene expression, such as those used in the DESeq2 and TMM, rely on other assumptions. For example, DESeq2 assumes that fewer than half of genes in the data are differentially expressed between samples. Furthermore, DESeq2 assumes that count data is not sparse, and can compute cell scaling factors close to 0 or 1 when used with sparse data, reflective of a higher dropout rate and not technical variation in gene count detection. With scaling factors close to 1, DESeq2 normalization behaves similarly to no normalization. In TMM, one sample is chosen to be a reference, thereby also assuming that most measured genes are not differentially expressed between samples. This is often not the case in im-SRT gene panels as they are often selected by identifying differentially expressed genes from prior scRNAseq data. Although our simulated gene panels were intentionally skewed for demonstration purposes, we emphasize that the approach by which these genes were selected can be used to design gene panels for im-SRT experiments and that choosing genes to focus on tissue regions or cell-types of interest in general may result in unintentionally skewed gene panels.

Despite this, count-based normalization can still provide robust insights when gene panels are representative of most of the component cell types in the tissue being assayed. Indeed, several publicly available im-SRT datasets were generated with the intention of broad spatial characterization of tissue spatial architecture and localization of component cell types. These studies in turn use gene panels that are broadly representative of the gene expression of the tissue's component cell types are therefore less susceptible to region- or cell type-specific biases in downstream analysis resulting from gene count-based normalization. Furthermore, as indicated in our results, biases in downstream transcriptomic analyses, particularly for fold change direction, tend to affect gene expression trends of more modest magnitude. Therefore, we expect strong biological signals to remain consistent even when using count-based normalization with skewed gene panels. As investigators move beyond tissue cell type characterization and towards utilizing im-SRT to answer targeted, hypothesis-driven questions about specific biological processes or cell types, we anticipate that more targeted gene panels will be needed. In fact, specialized gene panels skewed towards specific cell subpopulations are already commercially available. In these settings, it will be vital to choose gene expression normalization methods that mitigate the effects of such skewed gene panels on downstream transcriptomic analyses.

2. Similarly, conclusions in this study were drawn from only brain cannot generate to other organs. We suggest at least 4 organ types should be considered.

We thank the reviewer for the recommendation to extend our analysis to other organ types. As recommended, the new data types we investigated included 3 additional organs: human breast cancer (10X Genomics Xenium), mouse kidney (seqFISH), human liver (CosMx). As detailed in the response to reviewer comment #1, we design region-specific gene panels to investigate the effects of different normalization procedures with gene panels that overrepresent certain tissue regions. Specifically, for the 10X Genomics Xenium data, we simulate gene panels specific to ductal carcinoma in situ and invasive carcinoma. For the seqFISH data, we simulate gene panels specific to the kidney cortex, medulla, and pelvis. For the CosMx data, we simulate gene panels specific to Zone 1 and Zone 3 of the liver. For each dataset, we also simulate a gene panel with randomly sampled genes, to investigate the impact of normalization procedures with similarly-sized but representative gene panels.

As detailed in the response to the reviewer comment #1, overall, we observe similar results across organ types. With simulated gene panels that overrepresent gene expression from specific tissue regions, using count-based normalization procedures such as library size, scTransform, DESeq2, or TMM, results in region-specific on normalization scaling factors, differential gene expression, and gene fold change analysis.

The generalization of our results with additional data from different organs provides further support for our recommendation for caution with smaller and less representative imSRT panels. Based on these new results, we expect to see region- or celltype-specific biases in gene expression magnitude normalization using count-based gene expression normalization with gene panels that overrepresent gene expression of a specific tissue region or celltype, irrespective of the tissue being profiled. We have elaborated on this further in the discussion, reproduced above, in the response to comment #1 for convenience.

3. The included normalization strategies are also limited, is there no other methods for count normalization for single cell or ST data?

We thank the reviewer for the suggestion to include a count normalization method specifically developed for single cell data. We have now included an additional analysis using scTransform, which was developed specifically for count normalization for single-cell RNA sequencing data [6]. scTransform models gene counts using a generalized linear model with sequencing depth (i.e. library size) as the explanatory variable. The residuals of the resulting negative binomial regression are then treated as the normalized gene expression values. Because this normalization procedure is based on gene counts, we expect to see similar results to those we observed when using other count-based normalization methods such as library size, DESeq2, and TMM.

Indeed, we find that with gene panels that overrepresent gene expression of specific regions or cell types, normalization with scTransform results in region-specific biases in differential gene expression p-values, fold change analyses, and spatially variable gene expression, similar to those

observed with library size normalization. We note that scTransform does not use cell specific scaling factors and scaling factor correspondence was omitted for scTransform normalization. These effects are not observed with larger gene panels or when gene panels are representative of overall tissue gene expression. We observe these region-specific effects across data from different imSRT technologies and organ types as well as in our simulation with single-cell RNA sequence data from peripheral blood mononuclear cells.

Associated results are reproduced in the response to comment #1 above, with additional associated results reproduced below.

Revised text (Results lines 157-167):

Next, we performed gene count normalization with each of the normalization methods. We first assessed how consistent normalized gene expression magnitudes are when using the ventricle-skewed gene panel compared to normalized gene expression magnitudes with the full gene panel. To do this, we computed the correlation in normalized gene expression magnitudes between the skewed panel and the full gene panel (Figure 2C). Normalized gene expression magnitude correlations are the lowest for TMM normalization, ranging from around 0.5 to 0.98 for different genes. Library size and scTransform normalization resulted in similarly low correlations while DESeq2 resulted in comparatively higher correlations, all above 0.9. Volume normalization and no normalization resulted in perfectly correlated normalized gene expression magnitudes, as expected given that scaling factors are unchanged between different gene panels. We again find that these results are consistent for gene panels skewed towards other brain regions (Supp Fig 2E).

Revised text (Results lines 190-205):

Skewed gene panels with different normalization methods may lead to biases in differential gene expression results

Next, to evaluate the impact of these different normalization approaches with different gene panels on downstream analysis, we performed differential gene expression and gene fold change analysis. To do this, we compared the p-value of a Wilcoxon rank sum test of differential expression as well as the \log_2 fold change between cells in each brain region and all other regions based on normalized gene expression magnitudes achieved with the skewed gene panel versus the full gene panel for each normalization method. When using library size and scTransform normalization, p-values for ventricle-vs-all differential gene expression tests have less significant p-values with the ventricle-skewed gene panel compared to the full gene panel. In contrast, other differential gene expression test p-values are similar when using the ventricle-skewed gene panel and the full gene panel. This region-specific difference is less pronounced with TMM and DESeq2 normalization (Figure 3A). Note that with no normalization and volume normalization, the normalized gene expression magnitudes are the same (Figure 2) and thus downstream analysis results between the skewed gene panel and the full gene panel will be identical.

Revised text (Results lines 230-238):

With respect to gene expression fold changes, under library normalization and scTransform, \log_2 fold changes in ventricle-vs-all comparisons have lower \log_2 fold changes when using the ventricle-skewed gene panel compared to the full gene panel with some genes having negative \log_2 fold changes in the skewed gene panel and positive \log_2 fold changes in the full gene panel (switched negative). Conversely, \log_2 fold changes in comparisons for other regions (habenula-, fiber tract-, and dentate gyrus-vs-all) have more similar fold changes but include some genes that have a positive \log_2 fold change in the skewed gene panel and a negative \log_2 fold change in the full gene panel (switched positive). These region-specific effects are also seen with TMM normalization but are less pronounced and are not seen with DESeq2 normalization (Figure 3D).

Revised text (Results lines 259-263):

These examples show that the region-specific impacts on scaling factors when using region-skewed gene panels with count-based normalization can result in distortions in normalized gene expression distributions, which in turn lead to inconsistent fold change and differential gene expression results with different gene panels. Again, results are consistent for gene panels designed for other brain regions with the MERFISH data (Supp Fig 3-4).

Revised text (Results lines 292-323):

Skewed gene panels with different normalization methods may lead to false negatives in spatially variable gene expression analysis

To investigate the impact of different normalization methods with skewed gene panels on downstream analyses specific to SRT data, we sought to identify spatially variable genes (SVGs), genes with highly spatially correlated expression patterns. We used nnSVG with SEraster preprocessing to identify SVGs after gene count normalization with the ventricle-skewed gene panel and compared the results to those obtained with the full gene panel (Figure 5A). Comparing p-values with the ventricle-skewed gene panel to those with the full gene panel, we find that while there is some discordance in p-values without normalization and with cell volume normalization, likely reflective of the stochasticity of the nnSVG algorithm, this discordance is more pronounced with the evaluated count-based normalization methods. We quantify this effect by computing the false negative rate of SVG identification and we find higher false negative rates across count-based normalization methods with the ventricle-skewed gene panel (Figure 5B). As an example, we visualize the spatial gene expression pattern of one such false negative gene, *Pdgfra* (Figure 5C). Library size normalized *Pdgfra* expression with the ventricle-skewed gene panel does not show a discernible spatial pattern, compared to with the full gene panel, where we observe a region of high normalized *Pdgfra* expression. We observe similar results with gene panels designed for other brain regions with the MERFISH data (Supp Fig 9).

We further investigate the impact of different normalization methods with skewed gene panels on SVG analysis in im-SRT data of other tissues from different technologies. We observe similar effects in CosMx data of human liver with Zone 1-skewed gene panel. Comparing p-values with the Zone 1-skewed gene panel to those with the full gene panel, we find relatively higher levels of discordance with count-based normalization methods compared to with cell volume normalization or without normalization (Figure 5E). As with the ventricle-skewed gene panel with the MERFISH mouse brain im-SRT data, this effect is evident in the higher false negative rates of SVGs identified across count-based normalization methods, reaching up to 24%. We can see an example of this effect when looking at the spatial gene expression pattern of one false negative gene, *TNXB* (Figure 5F). Library size normalized *TNXB* expression with the Zone 1-skewed gene panel does not show a discernible spatial pattern, compared to with the full gene panel, where we observe several regions of high *TNXB* expression relative to the rest of the tissue.

Figure 2. Impact of region-skewed gene panel on normalization scaling factors and normalized gene expression magnitudes

A. For a coronal section of the mouse brain assayed by MERFISH, scatterplots of $\log_{10}(\text{scaling factors})$ for each cell based on different normalization methods (library size normalization, DESeq2 normalization, TMM normalization, and cell volume normalization) for the ventricle-skewed gene panel versus the full gene panel. Cells within the ventricle brain region are shown in purple. Cells in the fiber tract, habenula, and dentate gyrus brain regions are in grey. Red line indicates $x=y$.

- B.** Root mean squared error in scaling factors after each normalization method between the ventricle region-skewed gene panels and the full gene panel.
- C.** Boxplot of Pearson correlation coefficients (r) across genes for normalized gene expression magnitudes with the full gene panel versus the ventricle region-skewed gene panel across different normalization methods.
- D.** Scatterplot of normalized gene expression magnitudes when normalization was performed with the full gene panel versus the ventricle skewed gene panel for two highly expressed genes, *Lgr4* and *Htr2c*. Cells within the ventricle brain region are shown in purple. Cells in the fiber tract, habenula, and dentate gyrus regions are in grey. Red line indicates $x=y$.

Overall, we find that region- and celltype-specific biases when using count-based gene expression normalization procedures with skewed gene panels generalize across multiple commonly used count-based normalization procedures including library size, scTransform, DESeq2, and TMM. We note that these methods were developed for use with untargeted, full transcriptome data and therefore have underlying assumptions that are violated with im-SRT data, especially when gene panels are especially skewed. This indicates that this issue generalizes beyond the specific normalization procedure and further supports our recommendation for caution when using count-based normalization with smaller and less representative im-SRT panels. Based on these new results, we expect to see region- or celltype-specific biases in gene expression magnitude normalization using count-based gene expression normalization with gene panels that overrepresent gene expression of a specific tissue region or celltype, irrespective of the count-based normalization used.

We add the following text to the manuscript discussion:

Revised text (Discussion lines 405-431):

The gene count-based normalization methods evaluated in this work were developed for use with full-transcriptome RNA sequencing data and rely on assumptions that may be violated in targeted transcriptome profiling such as in im-SRT data, particularly with skewed gene panels. For example, library size normalization assumes that there are no genes that are only highly expressed in one subset of samples; that is, all samples have similar total gene expression. This assumption is inherently violated by designing gene panels to include canonical cell type markers for cell types of interest. scTransform models gene counts using a generalized linear model with sequencing depth (i.e. library size) as the explanatory variable to regress out the effect of sequencing depth. In this way, scTransform treats library size as a proxy for technical variables affecting gene count detection, such as RNA capture rate. In im-SRT, this may be confounded if gene count detection is additionally affected by differences in representation of different cell type gene expression profiles within measured genes. As such, we anticipate that scTransform could be modified to consider alternative or additional explanatory variables such as cell volume in the future. Other normalization methods that attempt to account for such compositional differences in gene expression, such as those used in the DESeq2 and TMM, rely on other assumptions. For example, DESeq2 assumes that fewer than half of genes in the data are differentially expressed between samples. Furthermore, DESeq2 assumes that count data is not sparse, and can compute cell scaling factors close to 0 or 1 when used with sparse data, reflective of a higher

dropout rate and not technical variation in gene count detection. With scaling factors close to 1, DESeq2 normalization behaves similarly to no normalization. In TMM, one sample is chosen to be a reference, thereby also assuming that most measured genes are not differentially expressed between samples. This is often not the case in im-SRT gene panels as they are often selected by identifying differentially expressed genes from prior scRNAseq data. Although our simulated gene panels were intentionally skewed for demonstration purposes, we emphasize that the approach by which these genes were selected can be used to design gene panels for im-SRT experiments and that choosing genes to focus on tissue regions or cell-types of interest in general may result in unintentionally skewed gene panels.

4. The author correctly stated that im-ST bulk the RNA molecules spots within cell regions to achieve single-cell resolution. But the cell-level expressions are fundamentally determined by cell segmentation algorithms and accuracy. It can also influence the cell size which was considered as a normalization factor by the authors. How does the cell segmentation algorithms and accuracy influence the performances of normalization strategies? Do different degrees of segmentation inaccuracies have large effect on the conclusions?

We thank the reviewer for raising this interesting point. We note that there are multiple upstream data acquisition and processing factors that may impact the analyses that we investigate in the present study. These steps include the sensitivity and specificity of molecule detection, error correction schemes, and as the reviewer notes, cell segmentation. We refer readers to an excellent comparative analysis of multiplexed in situ gene expression profiling technologies by Hartman and Satija, which investigates in detail the impact of these upstream factors [7].

Specifically, Hartman and Satija note the significant impact of segmentation approaches on the rate of non-specific molecule assignment to cells. More aggressive segmentation approaches that ensure that most detected molecules are assigned to cells lead to a higher rate of molecule misassignment compared to more conservative segmentation approaches. They additionally demonstrate that these misassignment errors can confound downstream spatial analyses.

We have now incorporated a citation to this work in our revised manuscript and add the following text to the discussion section in the manuscript:

Revised text (Discussion lines 478-484):

Still, the reliability of cell volume as a normalization factor is contingent upon the accuracy of cell segmentation. Segmentation approaches used vary substantially across studies and can result in segmented cell regions that are not representative of cell morphologies. Because cell segmentation remains a challenging problem, with novel experimental and computational approaches emerging rapidly, investigators using publicly available data with accessible cell volume or cell area estimates should therefore carefully assess the reliability of cell volume or cell area estimation methods, before using these data for normalization.

Revised text (Discussion lines 513-527):

Gene count normalization is an important step in the analysis of transcriptomics data as it precedes many downstream analyses. However, there are a number of upstream experimental and computational factors that may also impact the reliability of downstream transcriptomic analyses. These factors include molecule detection sensitivity and specificity, which can vary across im-SRT technologies depending on the molecule probe design, signal amplification approach, barcoding strategy used, uniformity of tissue permeability and perfusion, amongst other factors. Furthermore, cell segmentation continues to be a challenging step in im-SRT data analysis. Accuracy of cell segmentation directly impacts accuracy of molecule-to-cell assignment and, in turn, impacts sensitivity and specificity of cell count detection. This may present challenges, not only to the downstream transcriptomic analyses investigated in this study, but to other types of transcriptomic studies such as cell-cell communication analysis or analysis of impacts of cell neighborhood composition on gene expression. Additional investigation evaluating the impact of factors upstream of gene count normalization, including cell segmentation accuracy, as well as molecule detection sensitivity and specificity on in the analysis of im-SRT data is explored in Hartman and Satija.

5. The authors are also encouraged to consider more upstream factors that influence the gene expression quantification, for example the applied molecule spots detection methods and its parameters/accuracy.

We thank the reviewer for this suggestion. We agree that there are multiple upstream data acquisition and processing factors that may impact the downstream analyses that we investigate in this study. These steps include the sensitivity and specificity of molecule detection and error correction schemes.

Hartman and Satija [7] perform a comprehensive comparison of the sensitivity and specificity of imSRT technologies, including two of the datasets that we include in our study. Specifically, they perform a celltype-matched comparison of number of detected gene counts in mouse cortex imSRT data compared to mouse cortex single-cell RNA sequencing data to evaluate detection sensitivity. They additionally note the rate of detection of negative probes (probes that are not expected to generate a detectable signal) to assess detection specificity and find that detection rates for negative probes are 1-3 orders of magnitude lower than detection rates for real genes. Finally, as discussed in the response to reviewer comment #4, they note that molecule misassignment errors contribute significantly to non-specific gene expression in segmented cells and that this can confound downstream spatial analysis.

Again, we have now incorporated a citation to this work in our revised manuscript and add the text reproduced above in response to comment #4 to the discussion section in the manuscript **Revised text (Discussion lines 478-484):**

6. The MS majorly used scatter plots to evaluate different normalization strategies but lacks the use of scalar value metrics for a more straightforward message. We suggest using the

performance of downstream applications to improve this point. For example, the performance of spatial domain (<https://doi.org/10.1038/s41467-022-29439-6>), cell typing (<https://doi.org/10.1038/s41592-021-01264-7>), or spatially variable gene (<https://doi.org/10.1093/nar/gkad801>) etc.

We thank the reviewer for this suggestion. As suggested, we now include an analysis of spatially variable genes to demonstrate the impact of count-based normalization procedures with skewed gene panels. Specifically, with the simulated skewed MERFISH gene panels, we use nnSVG to identify spatially variable genes after normalization with each normalization method considered and compare p-values obtained when using the skewed panels to those obtained when using the full panel [8]. We find that with count-based normalization methods, while there is some degree on non-concordance between p-values after cell volume normalization or without normalization, likely due to stochasticity of the nnSVG algorithm, the degree on non-concordance after count-based normalization methods is consistently higher. This results in consistently higher false negative rates of spatially variable gene identification across count-based normalization methods when using skewed gene panels. We perform this analysis with the CosMx liver data and find similar results. Additionally, we visualize gene expression in tissue space for one example false negative gene for each of the MERFISH and CosMx datasets. In both examples, we identify hotspots of relatively higher gene expression with the full panel after library size normalization, which are not apparent with the skewed panel.

We note that in several of the data sets explored in this analysis, spatially variable gene analysis finds most genes to be spatially variable. This is likely due to the strong association between celltypes and tissue anatomical structures. This hinders our investigation of false positive rates since analysis of the full panel data does not yield non-significant genes to be treated as true negatives in our analysis.

We further add several scalar metrics to quantitatively compare between normalization methods and gene panels:

a. **Differential gene expression:** false positive and false negative error rate:

We compute error rates by comparing differential expression testing results with the skewed gene panels to the tests with the full panel using the same normalization procedure. Here, we consider the differential gene expression tests with the full gene panel to be ground truth.

We find that the quantitative error rate metrics reflect the qualitative observations made from the scatter plots comparing p-values with the skewed gene panels compared to the full gene panels. Across skewed gene panels, imSRT technologies, and organ types, we find that false positive rates and false negative rates are consistently higher with count-based normalization procedures. False positive and negative rates reach about 30% when using a kidney pelvis region-skewed gene panel in the mouse kidney seqFISH data after scTransform normalization.

b. **Gene expression fold change:** Switched positive and switched negative fold change direction error rate.

We note that the most serious downstream error in gene expression fold change analysis is one where the direction of the fold change does not reflect the true fold change direction in the cell subsets being analyzed. To this end, we compute the sign error rate in fold change analysis, where a switched positive indicates a gene whose true fold change is negative but is positive in our analysis within a skewed gene panel. A switched negative indicates the converse. Here, again, we take the result with the full gene panel to be the ground truth.

We find that the quantitative log₂ fold change direction error rate metrics reflect the qualitative observations made from the scatter plots comparing log₂ fold changes with the skewed gene panels compared to the full gene panels. Across skewed gene panels, imSRT technologies, and organ types, we find that switched positive rates and switched negative rates are consistently higher with count-based normalization procedures. Switched positive and negative rates reach more than 60% when using a liver Zone 1 region skewed gene panel in the human liver CosMx data after library size normalization.

c. Spatially variable gene identification: false negative rate

Here, we compute the false negative rate of spatially variable gene identification by comparing p-values obtained when using the skewed panels to those obtained when using the full panel. We find that with count-based normalization methods, while there is some degree on non-concordance between p-values after cell volume normalization or without normalization, likely due to stochasticity of the nnSVG algorithm, the degree on non-concordance after count-based normalization methods is consistently higher. This results in consistently higher false negative rates of spatially variable gene identification across count-based normalization methods when using skewed gene panels. This is observed in both the MERFISH and CosMx datasets. Again, we note that due to the high (often 100%) rate of significantly spatially variable genes with full gene panels in other datasets, we are unable to compute a false positive rate for spatially variable gene identification.

Associated results are reproduced in the responses to comments above (**Figures 2, 3, 4, 5, Supplementary Figures 5, 6, 7, 8**), with additional associated results reproduced below.

Revised text (Results lines 142-146):

We quantified the root mean squared error (RMSE) in scaling factors after using different normalization approaches with the ventricle-skewed gene panel compared to the full panel (Figure 2B). We find that these results are consistent for gene panels designed for other brain regions including the habenula, fiber tracts, and dentate gyrus (Supp Fig 2A-D).

Revised text (Results lines 226-228):

We quantify the rates of false positive and false negative differential expression test results (Figure 3C) and find up to a 13% error rate with multiple hypothesis correction depending on the count-based normalization approach used.

Revised text (Results lines 255-257):

We quantify the rates of switched positive and switched negative fold change results (Figure 3F) and find up to an 19% switch rate depending on the count-based normalization approach used.

Revised text (Results lines 273-283):

We observe similar region-specific effects on scaling factors, differential gene expression p-values, and gene \log_2 fold changes across im-SRT technologies and tissues. We quantify differential expression testing error rates and fold change switch rates (Figure 4). While error rates vary by dataset and simulated gene panel, we generally observe higher differential expression error rates with library size and scTransform normalization with skewed gene panels, reaching up to 17% with some simulated gene panels. Similarly, we generally observe higher fold change switch rates with library size and scTransform normalization reaching up to 32% with some simulated gene panels. We note that we observe higher rates of fold change switching at more modest fold changes compared to at higher fold changes where fold change directions tend to remain consistent between skewed gene panels and their corresponding full gene panel.

Revised text (Results lines 292-334):

Skewed gene panels with different normalization methods may lead to false negatives in spatially variable gene expression analysis

To investigate the impact of different normalization methods with skewed gene panels on downstream analyses specific to SRT data, we sought to identify spatially variable genes (SVGs), genes with highly spatially correlated expression patterns. We used nnSVG with SERaster preprocessing to identify SVGs after gene count normalization with the ventricle-skewed gene panel and compared the results to those obtained with the full gene panel (Figure 5A) [27], [28]. Comparing p-values with the ventricle-skewed gene panel to those with the full gene panel, we find that while there is some discordance in p-values without normalization and with cell volume normalization, likely reflective of the stochasticity of the nnSVG algorithm, this discordance is more pronounced with the evaluated count-based normalization methods. We quantify this effect by computing the false negative rate of SVG identification and we find higher false negative rates across count-based normalization methods with the ventricle-skewed gene panel (Figure 5B). As an example, we visualize the spatial gene expression pattern of one such false negative gene, *Pdgfra* (Figure 5C). Library size normalized *Pdgfra* expression with the ventricle-skewed gene panel does not show a discernible spatial pattern, compared to with the full gene panel, where we observe a region of high normalized *Pdgfra* expression. We observe similar results with gene panels designed for other brain regions with the MERFISH data (Supp Fig 9).

We further investigate the impact of different normalization methods with skewed gene panels on SVG analysis in im-SRT data of other tissues from different technologies. We observe similar effects in CosMx data of human liver with Zone 1-skewed gene panel.

Comparing p-values with the Zone 1-skewed gene panel to those with the full gene panel, we find relatively higher levels of discordance with count-based normalization methods compared to with cell volume normalization or without normalization (Figure 5E). As with the ventricle-skewed gene panel with the MERFISH mouse brain im-SRT data, this effect is evident in the higher false negative rates of SVGs identified across count-based normalization methods, reaching up to 24%. We can see an example of this effect when looking at the spatial gene expression pattern of one false negative gene, *TNXB* (Figure 5F). Library size normalized *TNXB* expression with the Zone 1-skewed gene panel does not show a discernible spatial pattern, compared to with the full gene panel, where we observe several regions of high *TNXB* expression relative to the rest of the tissue.

We note that in several of the data sets explored in this analysis, nnSVG analysis finds most genes to be spatially variable with the original gene panels. This is likely due to the spatial organization and variability of transcriptionally distinct cell-types. Since we treat genes with non-significant spatial variation with the full panel as true negatives in our error rate analysis, in these cases we find few or no true negatives and therefore omit quantification of false positive rates.

Overall, these results show that gene count-based normalization with skewed gene panels may result in unreliable results in downstream SVG analysis. Further, we observe these results across skewed gene panels and im-SRT datasets suggesting that these impacts on downstream SVG analyses can generalize to im-SRT data from different tissues assayed by different technologies.

Figure 5. Impact of region-skewed gene panel on spatially variable gene identification.

A. nnSVG significant spatially variable gene expression test $-\log_{10}(\text{p-values})$ adjusted for multiple hypothesis testing with normalized gene expression from the full gene panel versus the ventricle-skewed gene panel. Red line indicates $x=y$.

B. Significant spatially variable gene expression false negative rates after each normalization method, taking significant spatially variable gene expression results with the full gene panel to be ground truth.

C. Rasterized ($50 \mu\text{m}$) library size normalized gene expression magnitudes visualized in tissue space for *Pdgfra*, a representative spatial gene false negative, with the skewed gene panel (left) and the full gene panel (right).

D. Spatially variable gene identification in CosMx human liver im-SRT data with simulated Zone 1 region-skewed gene panel and full gene panel. nnSVG significant spatially variable gene expression test $-\log_{10}(\text{p-values})$ adjusted for multiple hypothesis testing with normalized gene expression from the full gene panel versus the Zone 1-skewed gene panel. Red line indicates $x=y$.

E. Significant spatially variable gene expression false negative rates after each normalization method, taking significant spatially variable gene expression results with the full gene panel to be ground truth.

F. Rasterized (50 μm) library size normalized gene expression magnitudes visualized in tissue space for *TNXB*, a representative spatial gene false negative, with the skewed gene panel (left) and the full gene panel (right).

7. The authors mentioned different gene panel selection methods, these panel selections should be considered, instead just using random selection or region-specific selection.

The reviewer is correct that there are many different ways to select genes in a targeted gene panel for imSRT. Previous works have focused on spatially profiling the organization of cell-types identified from dissociated scRNA-seq from the same tissue of interest. As such genes were selected based on differentially expressed marker genes among these identified cell populations [9], [10], [11]. Genes may also be selected based on prior biological knowledge regarding biological processes of interest and such approaches have been used previously to augment gene panels designed based on differentially expressed marker genes [12]. More recently, commercial vendors are now providing off-the-shelf gene panels that have been pre-designed to focus on specific biological processes [13] as well as fully customizable panels. As such, researchers may reasonably choose to design region specific or cell-type specific gene panels to focus on their tissue region or cell-type of interest, either based on prior biological knowledge or in a data-driven manner.

We have now clarified this in the main text:

Revised text (Discussion lines 433-448):

Despite this, count-based normalization can still provide robust insights when gene panels are representative of most of the component cell types in the tissue being assayed. Indeed, several publicly available im-SRT datasets were generated with the intention of broad spatial characterization of tissue spatial architecture and localization of component cell types. These studies in turn use gene panels that are broadly representative of the gene expression of the tissue's component cell types are therefore less susceptible to region- or cell type-specific biases in downstream analysis resulting from gene count-based normalization. Furthermore, as indicated in our results, biases in downstream transcriptomic analyses, particularly for fold change direction, tend to affect gene expression trends of more modest magnitude. Therefore, we expect strong biological signals to remain consistent even when using count-based normalization with skewed gene panels. As investigators move beyond tissue cell type characterization and towards utilizing im-SRT to answer targeted, hypothesis-driven questions about specific biological processes or cell types, we anticipate that more targeted gene panels will be needed. In fact, specialized gene panels skewed towards specific cell subpopulations are already commercially available. In these settings, it will be vital to choose gene expression normalization methods that mitigate the effects of such skewed gene panels on downstream transcriptomic analyses.

Revised text (Discussion lines 486-501):

In cases where cell volume or cell area data is not readily accessible or reliable, we recommend the use of non-spatially resolved single cell resolution, full transcriptome profiling approaches such as scRNA-seq data to inform gene panel design. This can be achieved by generating scRNA-seq and im-SRT datasets from the same tissue, with im-SRT panels designed based on differentially expressed genes identified from scRNA-seq data, as is demonstrated in several previous studies [14], [18], [31]. Furthermore, as collective atlasing efforts such as the Human BioMolecular Atlas Program (HuBMAP) [35], the Human Cell Atlas [36], and others continue to generate tissue-specific full-transcriptome gene expression data, we anticipate these resources may help provide a more untargeted view of gene expression across cell types in tissues of interest and enable researchers to design im-SRT gene panels that are representative of most of tissues' component cell types. These resources can also be used to evaluate gene panel skew when designing or using commercially available off-the-shelf gene panels prior to downstream gene expression normalization and transcriptomic analysis. In cases where gene panels are designed to be broadly representative of all cell-types within the tissue under investigation, we anticipate that single-cell gene count-based gene expression normalization approaches such as library size or scTransform normalization will be sufficiently reliable for reliable downstream analysis.

8. The major conclusions of this manuscript are (1) region specific gene panel negatively impact downstream analysis, and (2) more representative gene panel can mitigate some negative impact. These are within expectation. The manuscript would benefit from using more comprehensive datasets and metrics to derive a more useful guidelines for users or propose a computational solution or recommendations to address some limitations of current normalization methods. Otherwise, we cannot see direct contributions of this work. We think other methods evaluation papers on similar topics such as <https://doi.org/10.1038/s41467-022-35740-1> and <https://doi.org/10.1038/s41467-023-37168-7> should be helpful to improve this work.

We thank the reviewer for the suggestion. Based on the reviewer's suggestions and referenced methods evaluation papers we add the following to the manuscript and believe that it further highlights the direct contribution of our work to readers looking to both generate and analyze im-SRT data. We describe each of these contributions in more detail below:

- 1) We now introduce metrics to quantify errors in downstream analyses and highlight that in many cases, gene count-based normalization methods used with skewed gene panels can result in up to 30% false positive and false negative rates in differential expression testing, up to 60% fold change direction switches, and up to 24% false negative rates in spatially variable gene identification.
- 2) We introduce a novel method to quantify gene panel skew and guide readers on approaches to evaluate gene panel representativeness using scRNA-seq references.
- 3) We present step-by-step recommendations on gene count normalization choice and gene panel design to guide readers looking to evaluate gene panels in publicly available im-SRT data or design gene panels to mitigate downstream biases.

1) Error quantification metrics:

A. Differential gene expression error rates:

We compute error rates by comparing differential expression testing results with the skewed gene panels to the tests with the full panel using the same normalization procedure. Here, we consider the differential gene expression tests with the full gene panel to be ground truth, and compute false positive and false negative rates for the skewed gene panel accordingly.

We find that the quantitative error rate metrics reflect the qualitative observations made from the scatter plots comparing p-values with the skewed gene panels compared to the full gene panels. Across skewed gene panels, imSRT technologies, and organ types, we find that false positive rates and false negative rates are consistently higher with count-based normalization procedures. False positive and negative rates reach about 30% when using a kidney pelvis region-skewed gene panel in the mouse kidney seqFISH data after scTransform normalization.

B. Gene expression fold change error rates:

We note that the most serious downstream error in gene expression fold change analysis is one where the direction of the fold change does not reflect the true fold change direction in the cell subsets being analyzed. To this end, we compute the sign error rate in fold change analysis, where a switched positive indicates a gene whose true fold change is negative but is positive in our analysis within a skewed gene panel. A switched negative indicates the converse. Here, again, we take the result with the full gene panel to be the ground truth.

We find that the quantitative log₂ fold change direction error rate metrics reflect the qualitative observations made from the scatter plots comparing log₂ fold changes with the skewed gene panels compared to the full gene panels. Across skewed gene panels, imSRT technologies, and organ types, we find that switched positive rates and switched negative rates are consistently higher with count-based normalization procedures. Switched positive and negative rates reach more than 60% when using a liver Zone 1 region skewed gene panel in the human liver CosMx data after library size normalization.

C. Spatially variable gene identification error rates:

Here, we compute the false negative rate of spatially variable gene identification by comparing p-values obtained when using the skewed panels to those obtained when using the full panel. We find that with count-based normalization methods, while there is some degree on non-concordance between p-values after cell volume normalization or without normalization, likely due to stochasticity of the nnSVG algorithm, the degree on non-concordance after count-based normalization methods is consistently higher. This results in consistently higher false negative rates of spatially variable gene identification across count-based normalization methods when using skewed gene panels. This is observed in both the MERFISH and CosMx datasets. Again, we note that due to the high (often 100%) rate of significantly spatially variable genes with full gene panels in other datasets, we are unable to compute a false positive rate for spatially variable gene identification.

2) Gene panel skew quantification:

We introduce a metric to quantify gene panel skew toward tissue regions or celltypes. This metric uses Kullback-Leibler divergence to quantify the distance between observed counts per region or cell type in each panel and expected counts given cell region or cell type proportions in the data.

We find that this quantification is generally consistent with qualitative assessments of gene panel skew from gene expression heatmaps, with skewed gene panels from im-SRT data consistently having a larger skew compared to their full panel counterparts. Further, quantifying gene panel skew of gene panels of increasing size demonstrates decreased skew as gene panel size increases. This is expected given our simulation scheme and is consistent with the observed lower magnitude impacts on downstream differential expression and fold change analysis with gene panels of increasing size.

3) Step-by-step recommendations for normalization:

We also now add to the discussion recommendations on gene count normalization choice and gene panel design to guide readers looking to evaluate gene panels in publicly available im-SRT data or design gene panels to mitigate downstream biases.

Briefly, we recommend based on our simulation results, that when feasible, cell volume normalization should be used to account for differences in cell volume captured by imaging. If not feasible, we recommend using a single-cell count-based normalization method (library size or scTransform) for normalization if the gene panel is not skewed. Gene panel skew can be assessed using the metric we present and in conjunction with analysis of untargeted, whole transcriptome gene expression data from an appropriate reference, such as a scRNA-seq atlas, or from scRNA-seq data generated from the same tissue sample. Finally, if the im-SRT data at hand uses a skewed gene panel, we still recommend utilizing a gene count-based normalization approach but additionally, we recommend proceeding with caution when interpreting results of downstream analyses. For example, we recommend refraining for over-interpretation of small magnitude effects, even if statistically significant, as these may be artefacts of a skewed gene panel with gene count-based normalization.

Associated results are reproduced in response to comments #1 and #6 (**Figures 2-5, 7 Supplementary Figures 1, 5-8, 11**) with additional associated results reproduced below for convenience:

Revised text (Results lines 113-118):

We quantify gene panel skewness using a Kullback-Leibler divergence-based metric to confirm that the original full gene panel as well as random 100-gene gene panel are more representative of all brain regions compared to simulated region-skewed gene panels (Supp Fig 1F-G, methods). For both the original full gene panel and the region-skewed panel, we compared library size, scTransform, DESeq2, TMM, and cell volume normalization, in addition to no normalization.

Revised text (Discussion lines 464-511):

To this end, we provide a series of recommendations to guide the choice of normalization for im-SRT data (Figure 7). We recommend, as first choice, cell volume normalization to account for such partial cell capture within imaged regions. Cell area may serve as a proxy where volume estimates are not available. Our results further show that these normalization approaches may be preferred when working with skewed gene panels as scaling factors are independent of gene counts and as such gene panel differences do not impact downstream differential expression, fold change, or spatial gene expression variation analyses. Notably and as seen in the publicly available im-SRT datasets used in this work, cell volume data is not always readily available (Figure 4). However, based on the results in this study, we anticipate that normalization of gene expression magnitudes by cell volume to account for variation in captured cell volume will emerge as an especially important approach to correcting for within-experiment technical variation in a reliable manner. We therefore encourage investigators and commercial providers of im-SRT technologies to make cell volume estimates readily accessible.

Still, the reliability of cell volume as a normalization factor is contingent upon the accuracy of cell segmentation. Segmentation approaches used vary substantially across studies, and can result in segmented cell regions that are not representative of cell morphologies. Because cell segmentation remains a challenging problem, with novel experimental and computational approaches emerging rapidly, investigators using publicly available data with accessible cell volume or cell area estimates should therefore carefully assess the reliability of cell volume or cell area estimation methods, before using these data for normalization.

In cases where cell volume or cell area data is not readily accessible or reliable, we recommend the use of non-spatially resolved single cell resolution, full transcriptome profiling approaches such as scRNA-seq data to inform gene panel design. This can be achieved by generating scRNA-seq and im-SRT datasets from the same tissue, with im-SRT panels designed based on differentially expressed genes identified from scRNA-seq data, as is demonstrated in several previous studies. Furthermore, as collective atlasing efforts such as the Human BioMolecular Atlas Program (HuBMAP), the Human Cell Atlas, and others continue to generate tissue-specific full-transcriptome gene expression data, we anticipate these resources may help provide a more untargeted view of gene expression across cell types in tissues of interest and enable researchers to design im-SRT gene panels that are representative of most of tissues' component cell types. These resources can also be used to evaluate gene panel skew when designing or using commercially available off-the-shelf gene panels prior to downstream gene expression normalization and transcriptomic analysis. In cases where gene panels are designed to be broadly representative of all cell-types within the tissue under investigation, we anticipate that single-cell gene count-based gene expression normalization approaches such as library size or scTransform normalization will be sufficiently reliable for reliable downstream analysis.

Finally, in cases where gene panels are skewed towards a tissue region or cell type of interest, but cell volume or cell area information is unavailable, we still recommend utilizing

a gene count-based normalization approach to account for variation in captured cell volume. However, we recommend interpreting results of downstream analyses with caution. As shown in our results, while larger magnitude gene expression effects tend to remain evident with different gene panels, smaller effects are more susceptible to fold change direction changes while still remaining significant on differential expression testing. For this reason, we caution against over-interpreting biological conclusions from effects with small magnitude gene expression changes, and recommend further investigation of candidate effects.

Supplementary Figure 1. Gene expression of brain anatomical regions in full and simulated MERFISH gene panel

A. Heatmap showing scaled mean gene expression of 481 genes in full MERFISH panel for the four analyzed brain regions.

B. Heatmap showing scaled mean gene expression of the 100 genes in 100-gene gene panel skewed towards ventricle brain region for the four analyzed brain regions.

- C. Heatmap showing scaled mean gene expression of the 100 genes in 100-gene gene panel skewed towards habenula brain region for the four analyzed brain regions.
 - D. Heatmap showing scaled mean gene expression of the 100 genes in 100-gene gene panel skewed towards fiber tract brain regions for the four analyzed brain regions.
 - E. Heatmap showing scaled mean gene expression of the 100 genes in 100-gene gene panel skewed towards dentate gyrus brain regions for the four analyzed brain regions.
 - F. Heatmap showing scaled mean gene expression of the 100 genes in 1100-gene non-skewed gene panel for the four analyzed brain regions.
 - G. Gene panel skew of all simulated and original full MERFISH gene panels.
- In all panels, purple, green, blue, and red indicate the ventricles, habenula, fiber tracts, and dentate gyrus regions, respectively.

Supplementary Figure 11. Quantification of gene panel skew for monocyte-skewed gene panels of increasing size simulated from single-cell RNA sequencing data from peripheral blood mononuclear cells

Figure 7. Decision tree summarizing recommendations for gene count normalization method selection for im-SRT data

Reviewer #2:

In this manuscript, Atta et al. study the effects of various normalization methods applied to imaging-based spatial transcriptomics data. These techniques are currently targeting only a "small" set of selected genes and the authors show that if these genes are biased in the proportion of markers for distinct cell types, common ("library size", DESeq2 and edgeR) normalization scaling factors are consequently biased which impacts downstream differential analysis of gene expression visualization. The authors also make very clear that cell volume-based normalization is not affected by the composition of the gene panel (since it is based on the geometry of the cell instead of its observed expression data). Using simulations, they show that this effect is particularly pronounced on small gene panels, which corresponds to the current state of the technology.

1. The paper is well written and easy to follow. I do not have technical issues with the data presented. I however feel that while the authors make very clear that volume-based normalization is not affected by gene panel composition, unlike other normalization methods based on expression data, they do not address whether volume-based normalization is practically better for downstream analyses such as differential expression analysis. I therefore question whether the mention of volume-based normalization and its inclusion in the figures is really warranted without a comparison of volume-based normalization with other normalization methods on downstream analysis of actual data.

We thank the reviewer for the helpful suggestion. We agree that the inclusion of data for cell volume normalization is redundant after demonstrating cell volume scaling factors are not impacted gene panel composition and in turn, downstream analysis results are not impacted. We have therefore revised figures accordingly to only highlight the consistency of cell volume scaling factors in Figure 2.

However, we highlight cell volume normalization as an important alternative approach to gene count-based normalization. Such normalization may be necessary to correct for technical factors, namely partial cell volume capture in imaging. To this end, we have now included results from a simulation study demonstrating the potential confounding of downstream differential expression analysis resulting from partial cell volume capture. We have also now included further guidance to readers in the Discussion section on how to approach selection of normalization methods.

We reproduce the relevant text and results below for convenience:

Revised text (Discussion lines 450-511):

In general, count normalization methods have been developed to account for variation in detected gene counts across cells due to technical factors. One of these technical factors that has motivated the development of previous methods is variation in RNA capture rate in scRNA-seq. In im-SRT, we expect RNA capture rate to be similar across cells in the same experiment, provided that the entire cell volume is captured in the imaged region. For cells that lie at the boundaries

of the imaged regions, not all the cell volume will be captured resulting in a smaller gene count that is reflective of technical factors rather than biological ones. Similarly, for large cells that are not fully captured in imaged regions, cell orientation with respect to planes of imaging, can impact proportion of cell volume captured and in turn number of detected gene counts. We show in simulated gene expression data that when detected gene count differences due to partial volume capture compose a large proportion of total gene expression variation, gene expression normalization within individual im-SRT experiments may be needed to allow for reliable downstream analysis (Supp Fig 12).

To this end, we provide a series of recommendations to guide the choice of normalization for im-SRT data (Figure 7). We recommend, as first choice, cell volume normalization to account for such partial cell capture within imaged regions. Cell area may serve as a proxy where volume estimates are not available. Our results further show that these normalization approaches may be preferred when working with skewed gene panels as scaling factors are independent of gene counts and as such gene panel differences do not impact downstream differential expression, fold change, or spatial gene expression variation analyses. Notably and as seen in the publicly available im-SRT datasets used in this work, cell volume data is not always readily available (Figure 4). However, based on the results in this study, we anticipate that normalization of gene expression magnitudes by cell volume to account for variation in captured cell volume will emerge as an especially important approach to correcting for within-experiment technical variation in a reliable manner. We therefore encourage investigators and commercial providers of im-SRT technologies to make cell volume estimates readily accessible.

Still, the reliability of cell volume as a normalization factor is contingent upon the accuracy of cell segmentation. Segmentation approaches used vary substantially across studies and can result in segmented cell regions that are not representative of cell morphologies. Because cell segmentation remains a challenging problem, with novel experimental and computational approaches emerging rapidly investigators using publicly available data with accessible cell volume or cell area estimates should therefore carefully assess the reliability of cell volume or cell area estimation methods, before using these data for normalization.

In cases where cell volume or cell area data is not readily accessible or reliable, we recommend the use of non-spatially resolved single cell resolution, full transcriptome profiling approaches such as scRNA-seq data to inform gene panel design. This can be achieved by generating scRNA-seq and im-SRT datasets from the same tissue, with im-SRT panels designed based on differentially expressed genes identified from scRNA-seq data, as is demonstrated in several previous studies. Furthermore, as collective atlasing efforts such as the Human BioMolecular Atlas Program (HuBMAP), the Human Cell Atlas, and others continue to generate tissue-specific full-transcriptome gene expression data, we anticipate these resources may help provide a more untargeted view of gene expression across cell types in tissues of interest and enable researchers to design im-SRT gene panels that are representative of most of tissues' component cell types. These resources can also be used to evaluate gene panel skew when designing or using commercially available off-the-shelf gene panels prior to downstream gene expression normalization and transcriptomic analysis. In cases where gene panels are designed to be broadly

representative of all cell-types within the tissue under investigation, we anticipate that single-cell gene count-based gene expression normalization approaches such as library size or scTransform normalization will be sufficiently reliable for reliable downstream analysis.

Finally, in cases where gene panels are skewed towards a tissue region or cell type of interest, but cell volume or cell area information is unavailable, we still recommend utilizing a gene count-based normalization approach to account for variation in captured cell volume. However, we recommend interpreting results of downstream analyses with caution. As shown in our results, while larger magnitude gene expression effects tend to remain evident with different gene panels, smaller effects are more susceptible to fold change direction changes while still remaining significant on differential expression testing. For this reason, we caution against over-interpreting biological conclusions from effects with small magnitude gene expression changes, and recommend further investigation of candidate effects.

Supplementary Figure 12. Simulation of impact of partial cell volume capture on differential expression testing after gene expression normalization

A. Spatial position of two simulated cell subpopulations. Cell subpopulation A on average is positioned within the imaged Z planes (grey lines). Cell subpopulation B on average is positioned above the imaged Z planes.

B. Principal components of ground truth gene expression profiles of cell subpopulations A and B simulated using the splatter package with parameters estimated from the MERFISH mouse brain datasets. Gene expression profiles for cell subpopulations A and B are simulated with identical parameters.

- C. P-values for differential gene expression testing between cell subpopulation A and cell subpopulation B. P-values are treated as ground truth for later evaluation.
- D. Proportion of cell volume captured in imaged region based on simulated cell Z position.
- E. Total cell gene counts captured based on proportion of cell volume captured.
- F. Principal components of captured gene expression profiles of cell subpopulations A and B taking into account partial cell volume capture.
- G. P-values for differential gene expression testing with simulated captured gene counts between cell subpopulation A and cell subpopulation B without normalization (left) and after library size, DESeq2, and cell volume normalization (right), compared to ground truth p-values.

Figure 7. Decision tree summarizing recommendations for gene count normalization method selection for im-SRT data

Minor comments:

2. In scatter plots, it would be nice to a visual legend indicating that purple/green dots come from the region enriched in differentially-expressed genes for that region

We thank the reviewer for the recommendation. We now add to the figure captions additional text clarifying the colors used in the differential expression and fold change correspondence figures. We have modified all figures and include Figure 3 below as a representative example for convenience:

Figure 3. Impact of region-skewed gene panel on differential expression analysis and fold change evaluation

A. Wilcoxon rank sum test $-\log_{10}(\text{p-values})$ adjusted for multiple hypothesis testing for one region-vs-all differential gene expression test with normalized gene expression from the full gene panel versus the ventricle-skewed gene panel. Red line indicates $x=y$. Inset shows smaller $-\log_{10}(\text{adjusted p-values})$. Purple indicates ventricle-vs-all tests, grey indicates other region-vs-all tests.

B. Library size normalized gene expression magnitudes in cells in region of interest vs cells in other regions with the full gene panel and the ventricle skewed gene panel for the false positive (*Lgr4*, top) and false negative (*Ryk*, bottom) genes highlighted in A.

C. Differential gene expression false positive (top) and false negative (bottom) rates after each normalization method, taking differential gene expression results with the full gene panel to be ground truth.

D. One region-vs-all gene fold change with normalized gene expression from the full gene panel and the ventricle gene panel. Black lines indicate 0 \log_2 fold change and grey lines indicate 0.5 \log_2 fold change. Purple indicates ventricle-vs-all comparisons, grey indicates other region-vs-all comparisons.

E. Library size normalized gene expression magnitudes in cells in region of interest vs cells in other regions with the full gene panel and the ventricle skewed gene panel for the fold change switched positive (*Htr2c*, top) and switched negative (*Gpr75*, bottom) genes highlighted in D.

F. Gene expression fold change switched positive (top) and switched negative (bottom) rates after each normalization method, taking fold change results with the full gene panel to be ground truth.

3. Line 388: typo ("NA' and is" instead of "and NA' is")

We thank the reviewer for the note. We have corrected the error.

4. Simulation of skewed gene panels in im-SRT data : it could be interesting to generate random panels with various proportions of DEGs to see the impact of this proportion differential expression

We thank the reviewer for the interesting analysis suggestion. As suggested, we perform a simulation with the single-cell RNA sequencing data from peripheral blood mononuclear cells. We generate 100-gene gene panels with varying proportions of genes differentially overexpressed in monocytes (Reviewer Figure 1, below).

As the proportion of monocyte DE genes included increased, we find an increased cell type-specific impact on library size scaling factors and a corresponding increasing impact on cell type-specific bias in downstream differential gene expression p-values. These results reflect similar trends to those observed with monocyte-specific gene panels of increasing size in the simulated presented in Figure 6.

Reviewer Figure 1. Impact of increasing proportion of monocyte DE genes in a 100-gene gene panel on scaling factor and differential expression after library size normalization

A. Gene expression of peripheral blood mononuclear cell celltypes in gene panels simulated from scRNA-seq data to include increasing proportions of monocyte DE genes.

B. Scaling factors with monocyte-skewed gene panels with different proportions of monocyte DE genes.

C. Wilcoxon rank sum test $-\log_{10}(\text{p-values})$ adjusted for multiple hypothesis testing for one cell-type-vs-all differential gene expression tests with monocyte-skewed gene panels with different proportions of monocyte DE genes.

References:

- [1] “High resolution mapping of the breast cancer tumor microenvironment using integrated single cell, spatial and in situ analysis of FFPE tissue,” 10x Genomics. Accessed: Feb. 27, 2024. [Online]. Available: <https://www.10xgenomics.com/products/xenium-in-situ/preview-dataset-human-breast>
- [2] “CosMx SMI Human Liver FFPE Dataset - Seurat Object (no transcripts) | NanoString.” Accessed: Feb. 27, 2024. [Online]. Available: <https://nanosting.com/resources/seurat-object-no-transcripts-cosmx-smi-human-liver-ffpe-dataset/>
- [3] “seqFISH Mouse Kidney,” Spatial Genomics. Accessed: Feb. 27, 2024. [Online]. Available: <https://spatialgenomics.com/data/>
- [4] C. Evans, J. Hardin, and D. M. Stoebel, “Selecting between-sample RNA-Seq normalization methods from the perspective of their assumptions,” *Briefings in Bioinformatics*, vol. 19, no. 5, pp. 776–792, Sep. 2018, doi: 10.1093/bib/bbx008.
- [5] M.-A. Dillies *et al.*, “A comprehensive evaluation of normalization methods for illumina high-throughput rna sequencing data analysis,” *Briefings in Bioinformatics*, vol. 14, no. 6, pp. 671–683, Nov. 2013, doi: 10.1093/bib/bbs046.
- [6] C. Hafemeister and R. Satija, “Normalization and variance stabilization of single-cell RNA-seq data using regularized negative binomial regression,” *Genome Biology*, vol. 20, no. 1, p. 296, Dec. 2019, doi: 10.1186/s13059-019-1874-1.
- [7] A. Hartman and R. Satija, “Comparative analysis of multiplexed in situ gene expression profiling technologies.” bioRxiv, p. 2024.01.11.575135, Jan. 24, 2024. doi: 10.1101/2024.01.11.575135.
- [8] “nnSVG for the scalable identification of spatially variable genes using nearest-neighbor Gaussian processes | Nature Communications.” Accessed: Feb. 27, 2024. [Online]. Available: <https://www.nature.com/articles/s41467-023-39748-z>
- [9] R. Chen *et al.*, “Decoding molecular and cellular heterogeneity of mouse nucleus accumbens,” *Nat Neurosci*, vol. 24, no. 12, Art. no. 12, Dec. 2021, doi: 10.1038/s41593-021-00938-x.
- [10] T. Lohoff *et al.*, “Integration of spatial and single-cell transcriptomic data elucidates mouse organogenesis,” *Nat Biotechnol*, vol. 40, no. 1, Art. no. 1, Jan. 2022, doi: 10.1038/s41587-021-01006-2.
- [11] B. Zhang *et al.*, “A human embryonic limb cell atlas resolved in space and time,” *Nature*, pp. 1–11, Dec. 2023, doi: 10.1038/s41586-023-06806-x.
- [12] J. R. Moffitt *et al.*, “Molecular, spatial, and functional single-cell profiling of the hypothalamic preoptic region,” *Science*, vol. 362, no. 6416, Nov. 2018, doi: 10.1126/science.aau5324.
- [13] “Pre-designed Xenium Gene Expression Panels - Official 10x Genomics Support,” 10x Genomics. Accessed: Jul. 25, 2023. [Online]. Available: <https://www.10xgenomics.com/support/in-situ-gene-expression/documentation/steps/panel-design/pre-designed-xenium-gene-expression-panels>

Second round of review

Reviewer 1

The author addressed some of my previous concerns, but other major concerns have not been properly addressed, especially comment 4.

The datasets used in this study were collected from different laboratories and therefore processed using different upstream tools and configurations, for example, segmentation and spot detection, making the quantification of gene expression per cell highly uncontrolled. This flaw may affect the reliability of the conclusions drawn from each of the data. This concern cannot simply be responded to by referring to other work, but rather by conducting real experiments. Otherwise, how can you confirm that the results of this analysis are not produced by irrelevant factors introduced by uncontrolled upstream analysis? We believe at least the authors should use consistent spot detection or segmentation protocols on these data to increase the validity of this study.

Gene count normalization in single-cell imaging-based spatially resolved transcriptomics: Point-by-point Reviewer Response

Overview:

We thank the editor and the reviewer for their feedback. We have now revised the manuscript to address the additional requested points of clarification. Below are the corresponding point-by-point responses to the reviewer comments. Throughout this point-by-point response, reviewer comments are shown in **blue**, with our responses in **green**, and changes to the manuscript in **black**.

Reviewer #1:

The author addressed some of my previous concerns, but other major concerns have not been properly addressed, especially comment 4.

The datasets used in this study were collected from different laboratories and therefore processed using different upstream tools and configurations, for example, segmentation and spot detection, making the quantification of gene expression per cell highly uncontrolled. This flaw may affect the reliability of the conclusions drawn from each of the data. This concern cannot simply be responded to by referring to other work, but rather by conducting real experiments. Otherwise, how can you confirm that the results of this analysis are not produced by irrelevant factors introduced by uncontrolled upstream analysis? We believe at least the authors should use consistent spot detection or segmentation protocols on these data to increase the validity of this study.

We agree with the reviewer that upstream platform-specific technical factors can present several additional sources of variability between im-SRT experiments. Other studies have highlighted using one-to-one comparisons of im-SRT data generated from the same laboratory from technical replicates of the same biological sample that there are differences in probe detection rates, sensitivity and specificity of probe assignment to transcript species, as well as differences in the accuracy of default segmentation between platforms, impacting the generalizability of results from im-SRT experiments and may affect the reliability of biological conclusions drawn from those results [1], [2], [3]. We have now included additional references to these other works to refer readers to more detailed resources on how upstream factors may influence biological interpretation.

However, in our study, we are focusing on demonstrating that independent of these upstream platform-specific technical factors, experimental design choices - namely gene panel composition and count normalization method - present additional sources of variability and can introduce systematic errors affecting downstream analysis. We have replicated our observations across many tissues, gene panels, and technologies to demonstrate that regardless of the variation in spot detection, segmentation, and other upstream factors, conclusions generalize across a variety of contexts.

We emphasize that in our analyses, error rates are derived by comparing between gene panels simulated from data from the same experiment, thereby controlling for inter-experiment technical differences. These analyses demonstrate that gene panel composition and count normalization method can introduce systematic errors in downstream analysis even in within-experiment comparisons (where all upstream factors are the same). We have now re-emphasized this in our discussion (line 529):

Still, independent of these upstream technology-specific technical factors, experimental design choices - namely gene panel composition and count normalization method - present additional sources of variability and can introduce systematic biases affecting downstream analysis (Supplementary Note 1).

In addition, we emphasize that our conclusion that gene panel composition and count normalization method present additional sources of variability and can introduce systematic errors affecting downstream analysis can be demonstrated theoretically, as delineated in a new supplementary note (Supplementary Note 1), provided below. We further include a graphical representation of these theoretical results in Supplementary Figures 13 and 14, also reproduced below. We hope that this clarifies for the reviewer how our conclusion is not produced by irrelevant factors introduced by uncontrolled upstream analysis.

Supplementary Note 1:

The impact of gene panel composition and count normalization method on downstream differential expression analysis can be demonstrated mathematically using the theoretical example below.

Suppose we are measuring gene expression in a tissue composed of two cell-types, A and B, with the following gene expression matrix:

$$G = \begin{pmatrix} x_A & x_B \\ y_A & y_B \\ z_A & z_B \end{pmatrix}$$

where the columns of G denote the gene expression profiles of cell-types A and B, respectively, and where x , y , and z , denote the gene counts of groups of genes in cells of individual cell-types. Additionally, suppose that cells of cell-type A and cell-type B have similar cell volumes v .

If we consider gene expression data from 3 cells, A , A' , B , where cells A and A' are of cell-type A, and cell B is of cell-type B, and cell A' is partially captured with half of its cell volume imaged. The detected count matrix G will be:

$$G = \begin{pmatrix} x_A & \frac{1}{2}x_A & x_B \\ y_A & \frac{1}{2}y_A & y_B \\ z_A & \frac{1}{2}z_A & z_B \end{pmatrix}$$

where the columns of G are the detected counts for cells A , A' , and B , respectively, and the imaged cell volumes, V , will be:

$$V = \begin{pmatrix} v & \frac{1}{2}v & v \end{pmatrix}$$

Defining gene panel skew:

Gene panel skew, as defined in this study, can be determined by considering the relative number of counts detected for each cell-type. In a skewed gene panel, on average, total gene expression for cells of one cell-type is higher than that of other cell-types. In the above example, a gene panel is skewed if:

$$x_A + y_A + z_A \gg x_B + y_B + z_B$$

A skewed gene panel can include genes that are not differentially expressed, as well as differentially expressed genes that are markers for each cell-type (i.e. highly expressed in one cell-type and not others), provided that, on average, the total detected gene counts for cells of one cell-type is much greater than the total detected gene counts of cells of other cell-types.

Conversely, a gene panel is non-skewed if, on average, total detected gene counts are similar between cell-types. In the above example, a gene panel is non-skewed if:

$$x_A + y_A + z_A \approx x_B + y_B + z_B$$

Gene count normalization in a skewed gene panel (Supplementary Figure 13):

Consider a skewed gene panel where group x genes are not differentially expressed between cell-types, and group y and group z genes are specific marker genes for cell-types A and B, respectively, i.e.:

$$x_A \approx x_B \approx x,$$

$$y_A \gg y_B = 0,$$

$$z_B \gg z_A = 0,$$

$$y_A \gg z_B$$

Detected gene counts G_o for cells A, A', and B are:

$$G_o = \begin{pmatrix} x & \frac{1}{2}x & x \\ y_A & \frac{1}{2}y_A & 0 \\ 0 & 0 & z_B \end{pmatrix}$$

No gene count normalization:

Given these detected counts, if we were to compare gene expression for each pair of gene groups for each pair of cells (i.e. perform 1-vs-all differential gene expression) without any gene count normalization, we would erroneously conclude that:

- cell A overexpressed group x genes compared to cell A'
- cell B overexpressed group x genes compared to cell A'
- cell A overexpresses group y genes compared to cell A'

Library size normalization:

Given detected gene counts G_o , total detected counts (i.e. library size) for each cell is:

$$L = \begin{pmatrix} x + y_A & \frac{1}{2}(x + y_A) & x + z_B \end{pmatrix}$$

and library size normalized counts G_L are:

$$G_L = \begin{pmatrix} \frac{x}{x + y_A} & \frac{x}{x + y_A} & \frac{x}{x + z_B} \\ \frac{y_A}{x + y_A} & \frac{y_A}{x + y_A} & 0 \\ 0 & 0 & \frac{z_B}{x + z_B} \end{pmatrix}$$

As specified above, this gene panel is skewed towards group y genes, i.e. $y_A \gg z_B$ and therefore $\frac{x}{x + y_A} \ll \frac{x}{x + z_B}$. A 1-vs-all differential gene expression analysis of library size normalized counts between cells here would erroneously conclude that:

- cell B overexpressed group x genes compared to cell A
- cell B overexpressed group x genes compared to cell A'

Cell volume normalization:

Given detected gene counts G_o and cell volumes $V = (v \quad \frac{1}{2}v \quad v)$, cell volume normalized counts G_V are:

$$G_V = \begin{pmatrix} \frac{x}{v} & \frac{x}{\frac{1}{2}v} & \frac{x}{v} \\ \frac{y_A}{v} & \frac{y_A}{\frac{1}{2}v} & 0 \\ 0 & 0 & \frac{z_B}{v} \end{pmatrix}$$

Here, a 1-vs-all differential gene expression analysis of volume normalized gene counts would correctly conclude that:

- Group x genes are not differentially expressed between cells A , A' , and B
- Group y genes are not differentially expressed between cells A and A' , but are overexpressed in cells A and A' compared to cell B .
- Group z genes are overexpressed in cell B compared to cells A and A' .

Gene count normalization in a non-skewed gene panel (Supplementary Figure 14):

Consider a non-skewed gene panel where group x genes are not differentially expressed between cell-types, and group y and group z genes are specific marker genes for cell-types A and B, respectively, i.e.:

$$x_A \approx x_B \approx x,$$

$$y_A \gg y_B = 0,$$

$$z_B \gg z_A = 0,$$

$$y_A \approx z_B$$

Detected gene counts G_o for cells A, A', and B are:

$$G_o = \begin{pmatrix} x & \frac{1}{2}x & x \\ y_A & \frac{1}{2}y_A & 0 \\ 0 & 0 & z_B \end{pmatrix}$$

No gene count normalization:

As in the skewed gene panel case, not accounting for partial cell volume imaging and the resulting undersampling of detected gene counts results in false positive results on 1-vs-all differential gene expression where we would erroneously conclude that:

- cell A overexpressed group x genes compared to cell A'
- cell B overexpressed group x genes compared to cell A'
- cell A overexpresses group y genes compared to cell A'

Library size normalization:

As above, library size normalized counts G_L are:

$$G_L = \begin{pmatrix} \frac{x}{x + y_A} & \frac{x}{x + y_A} & \frac{x}{x + z_B} \\ \frac{y_A}{x + y_A} & \frac{y_A}{x + y_A} & 0 \\ 0 & 0 & \frac{z_B}{x + z_B} \end{pmatrix}$$

Here, $y_A \approx z_B$ and therefore $\frac{x}{x+y_A} \approx \frac{x}{x+z_B}$. A 1-vs-all differential gene expression analysis of library size normalized counts between cells here would correctly identify that conclude that:

- Group x genes are not differentially expressed between cells A , A' , and B
- Group y genes are not differentially expressed between cells A and A' , but are overexpressed in cells A and A' compared to cell B .
- Group z genes are overexpressed in cell B compared to cells A and A' .

Notably, when $y_A \approx z_B$, cell library size is proportional to imaged cell volume such that normalizing by cell library size correctly accounts for differences in sampling due to incomplete cell capture.

References:

- [1] A. Hartman and R. Satija, “Comparative analysis of multiplexed in situ gene expression profiling technologies.” bioRxiv, p. 2024.01.11.575135, Jan. 24, 2024. doi: 10.1101/2024.01.11.575135.
- [2] D. P. Cook *et al.*, “A Comparative Analysis of Imaging-Based Spatial Transcriptomics Platforms.” bioRxiv, p. 2023.12.13.571385, Dec. 14, 2023. doi: 10.1101/2023.12.13.571385.
- [3] A. Rademacher *et al.*, “Comparison of spatial transcriptomics technologies using tumor cryosections.” bioRxiv, p. 2024.04.03.586404, Apr. 05, 2024. doi: 10.1101/2024.04.03.586404.

Supplementary Figure 13. Impact of skewed gene panel and choice of normalization method.

A. Skewed gene panels overrepresent the gene expression of specific tissue cell-types or regions. In a tissue with two cell-types, 1 and 2, group X genes are not differentially expressed between cell-types, and group Y and group Z genes are overexpressed in cell-types 1 and 2, respectively. Group Y includes more/more highly expressed genes than group Z making the gene panel skewed towards cell-type A.

B. Partial imaging of cells in cell-types results in undersampling cell gene expression. Cells A and B are of cell-type 1, where cell A is fully imaged and cell B is partially imaged. Cell C is of cell-type 2 and is fully imaged.

C. True differential expression relationships between cells A, B, C, given their cell-types. Group X genes are not differentially expressed between cells. Group Y genes are overexpressed in cells A and B compared to cell C. Group Z genes are overexpressed in cell C compared to cells A and B.

D. Cell library size (i.e. total detected counts) based on measured genes and accounting for sampling differences due to proportion of cell volume imaged.

Bottom Left. Detected gene counts and differential expression without normalization. Group X genes are undercounted in cell B due to partial cell volume imaging resulting in false positive DE when comparing to cells A and C. Similarly, Group Y genes are undercounted in cell B due to partial cell volume imaging resulting in false positive DE when comparing to cell A.

Bottom Middle. Volume normalized gene counts and differential expression with volume normalization. Volume normalization accounts for undersampling due to partial cell volume imaging. DE gene groups are correctly identified.

Bottom Right. Library size normalized gene counts and differential expression with library size normalization. Library size normalization incorrectly accounts for partial cell volume imaging. Normalized group X gene counts are inflated for cell C compared to cells A and B. This results in false positive DE for group X genes in cell C when comparing to cells A and B.

Supplementary Figure 14. Impact of non-skewed gene panel and choice of normalization method.

A. In non-skewed gene panels, tissue cell-types or regions have similar total gene expression. In a tissue with two cell-types, 1 and 2, group X genes are not differentially expressed between cell-types, and group Y and group Z genes are overexpressed in cell-types 1 and 2, respectively. Group Y and group Z genes are expressed at similar magnitudes in their respective cell subpopulations making the gene panel non-skewed.

B. Partial imaging of cells in cell-types results in undersampling cell gene expression. Cells A and B are of cell-type 1, where cell A is fully imaged and cell B is partially imaged. Cell C is of cell-type 2 and is fully imaged.

C. True differential expression relationships between cells A, B, C, given their cell-types. Group X genes are not differentially expressed between cells. Group Y genes are overexpressed in cells A and B compared to cell C. Group Z genes are overexpressed in cell C compared to cells A and B.

D. Cell library size (i.e. total detected counts) based on measured genes and accounting for sampling differences due to proportion of cell volume imaged.

Bottom Left. Detected gene counts and differential expression without normalization. Group X genes are undercounted in cell B due to partial cell volume imaging resulting in false positive DE when comparing to cells A and C. Similarly, Group Y genes are undercounted in cell B due to partial cell volume imaging resulting in false positive DE when comparing to cell A.

Bottom Middle. Volume normalized gene counts and differential expression with volume normalization. Volume normalization accounts for undersampling due to partial cell volume imaging. DE gene groups are correctly identified.

Bottom Right. Library size normalized gene counts and differential expression with library size normalization. Library size normalization accounts for undersampling due to partial cell volume imaging. DE gene groups are correctly identified.

Third round of review

Reviewer 1

Given the authors' extensive work and the importance of the topic, I am ok for the publication of this manuscript. However, I still believe that controlling for irrelevant factors is necessary for such an analysis article, which may be left for future work to conduct a more rigorous analysis. Please discuss the potential improvement of the analysis by controlling segmentation or other factors (such as PMID: 38218939).